# Synaptic connectivity of sensorimotor circuits for vocal imitation in the songbird

Massimo Trusel*, Ziran Zhao, Danyal H Alam, Ethan S Marks, Maaya Z Ikeda, Todd F Roberts*

Department of Neuroscience, UT Southwestern Medical Center, Dallas, United States

## eLife Assessment

The songbird vocal motor nucleus HVC contains cells that project to the basal ganglia, the auditory system, or to downstream vocal motor structures. In this **fundamental** study, the authors conduct optogenetic circuit mapping to clarify how four distinct inputs to HVC act on these distinct HVC cell types. They provide **compelling** evidence that all long range projections engage inhibitory circuits in HVC and can also exhibit cell-type specific preferences in monosynaptic input strength. Understanding HVC at this microcircuit level is critical for constraining models of song learning and production.

*For correspondence:
Massimo.trusel@utsouthwestern.edu (MT);
todd.roberts@utsouthwestern.edu (TFR)

Competing interest: The authors declare that no competing interests exist.

**Abstract** Sensorimotor computations for learning and behavior rely on precise patterns of synaptic connectivity. Yet, we typically lack the synaptic wiring diagrams for long-range connections between sensory and motor circuits in the brain. Here, we provide the synaptic wiring diagram for sensorimotor circuits involved in learning and production of male zebra finch song, a natural and ethologically relevant behavior. We examined the functional synaptic connectivity from the 4 main sensory afferent pathways onto the three known classes of projection neurons of the song premotor cortical region HVC. Recordings from hundreds of identified projection neurons reveal rules for monosynaptic connectivity and the existence of polysynaptic ensembles of excitatory and inhibitory neuronal populations in HVC. Circuit tracing further identifies novel connections between HVC's presynaptic partners. Our results indicate a modular organization of ensemble-like networks for integrating long-range input with local circuits, providing important context for information flow and computations for learned vocal behavior.

## Introduction

Patterns of synaptic connectivity anchor how activity flows within sensory and motor networks in the brain. Consequently, these synaptic connections delimit and help define how sensory and premotor circuits interact. Mapping synaptic pathways between these circuits is therefore fundamental to understanding the neural computations involved in learning and producing behaviors. Great strides have been made in developing technologies for large-scale recording of neuronal activity across brain regions (*Steinmetz et al., 2021*; *Demas et al., 2021*; *Manley et al., 2024*) and in building functional connectomes of local (<1 mm) synaptic circuits (*Bae et al., 2021*). Less progress has been made at intermediate levels, which aim to build wiring diagrams for long-range synaptic connections in the brain (*Petreanu et al., 2009*). Moreover, while most evidence for long-range synaptic connectivity has been provided piecemeal, we now have the opportunity to systematically apply circuit dissection methods to other animal models in which we still know relatively little about patterns of long-range synaptic connectivity.

We were inspired to understand the wiring diagram for sensorimotor circuits critical for birdsong, a complex and volitionally produced skilled behavior that, like speech and language, is dependent on forebrain circuits for its fluent production (*Konopka and Roberts, 2016*; *Doupe and Kuhl, 1999*). The dedicated neural circuits associated with song provide a powerful model in which to study how synaptically linked sensory and motor networks of neurons control a complex behavior. The courtship song of male zebra finches is one of the better studied naturally learned behaviors (*Immelmann, 1969*; *Price, 1979*; *Sossinka and Bohner, 1980*; *Böhner, 1983*; *Zann, 1984*; *Clayton, 1987*; *Böhner, 1990*; *Zann, 1996*; *Tchernichovski et al., 1999*; *Tchernichovski et al., 2001*; *Funabiki and Konishi, 2003*; *Goller and Cooper, 2004*; *Alam et al., 2024*). Zebra finch song is controlled by a relatively discrete set of interconnected forebrain regions located in the dorsal ventricular ridge (DVR; *Nottebohm et al., 1976*; *Konishi, 1989*; *Schmidt and Goller, 2016*; *Schmidt et al., 2004*; *Mooney, 2009*; *Fee et al., 2004*; *Roberts et al., 2017*; *Aronov et al., 2008*; *Moll et al., 2023*; *Daliparthi et al., 2019*). The DVR is the avian analogue of the mammalian neocortex, and like the neocortex, it contains parcellated regions that (i) receive sensory information from the thalamus, (ii) regions that process information connected by mostly ipsilaterally projecting intratelencephalic projection neurons (PNs), and (iii) output motor circuits projecting back to the thalamus and to motor regions throughout the brainstem (*Stacho et al., 2020*; *Colquitt et al., 2021*). The vocal premotor nucleus HVC (proper name) is a central hub in the DVR song network. HVC is necessary for juvenile song learning and adult song production. It also serves as the primary synaptic interface between sensory pathways and the premotor circuits involved in learning and controlling singing behavior (*Nottebohm et al., 1976*; *Roberts et al., 2017*; *Aronov et al., 2008*; *Ikeda et al., 2020*; *Zhao et al., 2019*; *Bauer et al., 2008*; *Garcia-Oscos et al., 2021*).

Although anatomical evidence delineated the main input and output pathways of HVC decades ago (*Nottebohm et al., 1976*; *Roberts et al., 2017*; *Nottebohm et al., 1982*; *Vates et al., 1996*; *Akutagawa and Konishi, 2010*; *Roberts et al., 2008*), a cell-type-specific synaptic wiring diagram of this core circuitry has remained out of reach. HVC receives intratelencephalic input from at least four sensory and sensorimotor regions in the DVR (NIf, Av, mMAN, and RA) and one thalamic brain region, Uva (*Figure 1A and B*; see *Figure 1* legend for anatomical descriptions). Several lines of evidence support the idea that song learning and vocal motor control involve some of these pathways projecting into HVC (*Roberts et al., 2017*; *Zhao et al., 2019*; *Garcia-Oscos et al., 2021*; *Roberts et al., 2008*; *Foster and Bottjer, 2001*; *Danish et al., 2017*; *Coleman and Vu, 2005*; *Hamaguchi and Mooney, 2012*; *Hamaguchi et al., 2016*; *Roberts et al., 2012*). In addition, HVC has three distinct classes of intratelencephalic projecting neurons that function as the output pathways important for song learning and song motor control (*Roberts et al., 2017*). HVC's projection onto RA through $HVC_{RA}$ neurons forms the descending song motor pathway necessary for song production. HVC's projection onto Area X through $HVC_X$ neurons forms a DVR-striatal circuit that is important for song plasticity. HVC's projection onto Avalanche through $HVC_{Av}$ neurons forms the song circuit's projection back to the auditory DVR, a pathway that is important for evaluating motor performance during song learning (*Roberts et al., 2017*).

Mapping the synaptic connectivity between HVC's input and output pathways has been hindered by the lack of tools for robustly manipulating these circuits. We have overcome this issue by employing better-performing optogenetic channels packaged in an adeno-associated virus (AAV) that we optimized for expression throughout various portions of the song circuitry. Specifically, we made use of the axonal expression of the opsin eGtACR1 (*Mardinly et al., 2018*; *Govorunova et al., 2015*). GtACRs are blue light-driven Cl⁻ channels that, while hyperpolarizing at the soma and dendrites, are depolarizing at axon terminals due to the shifted internal Cl⁻ concentration in axonal compartments (*Malyshev et al., 2017*; *Messier et al., 2018*; *Khirug et al., 2008*). Here, we use viral expression of eGtACR1 to achieve independent control of synaptic release from NIf, Uva, mMAN, and Av. We paired this with whole-cell patch-clamp recordings from visually targeted HVC projection neurons (HVC-PNs; $HVC_X$, $HVC_{RA}$, $HVC_{Av}$), identified using retrograde tracer injections into each efferent region (*Figure 1C*). This permitted the first large-scale cell-type-specific synaptic connectivity mapping, using gold-standard electrophysiological methods, in songbirds. In ex vivo brain slices from adult male zebra finches, we mapped the properties of synaptic transmission at the 12 afferent-HVC-PN combinations. Via recordings from hundreds of HVC PNs, we provide a comprehensive polysynaptic and monosynaptic mapping of the connectivity between these afferent sensory pathways and the output

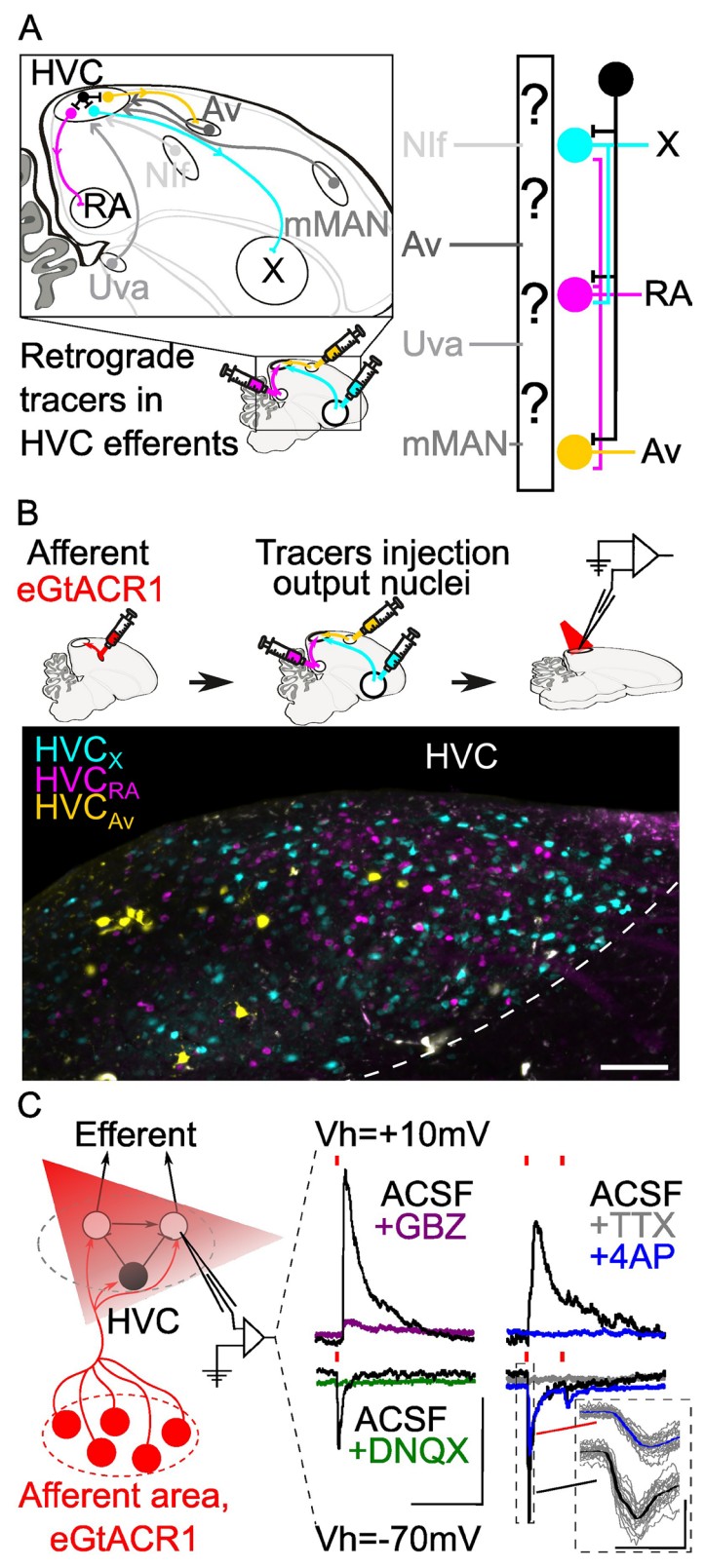

**Figure 1.** Strategy for opsin-assisted HVC circuit mapping. (**A**) (left) Parasagittal schematic of the zebra finch brain illustrating HVC afferents (grey) from nucleus interface of the nidopallium (NIf), nucleus avalanche (Av), medial magnocellular nucleus of the anterior nidopallium (mMAN) and nucleus Uvaeformis (Uva), and the three classes of HVC-PNs: HVC$_X$ (projecting to Area X, cyan), HVC$_{RA}$ (projecting to RA, magenta), and HVC$_{Av}$ (projecting

*Figure 1 continued on next page*

*Figure 1 continued*

to Av, yellow). (right) Schematic illustrating the known synaptic connectivity of HVC's input and output pathways (interneurons schematized in black). (**B**) (top) Schematic of the workflow including opsin expression in afferent areas, retrograde tracer injection in afferent areas and ,whole cell patch-clamp recording of light-evoked currents in acute brain slices. (bottom) Sample image of retrogradely labeled HVC-PN classes in a brain slice used for patch-clamp recordings (scalebar 100 μm). (**C**) Schematic of whole-cell recording of light-evoked synaptic currents in HVC-PNs receiving monosynaptic inputs from one of the afferent areas expressing eGTACR1 (red), as well as polysynaptic inputs from local HVC-PNs (white circle) and interneurons (black circle). Recordings are performed at holding potential (Vh)=+10 mV and –70 mV. Sample traces report the effect of bath application of DNQX (green) and gabazine (GBZ, purple), which suppress oEPSC and oIPSC, respectively. Glutamatergic monosynaptic currents are pharmacologically isolated by bath application of TTX +4AP: sample traces (average of 20 sweeps) portray a typical case of a cell displaying oEPSC with a monosynaptic component (inset shows 20 oEPSCs sweeps in grey, averaged in the thick traces in black and blue), and polysynaptic oIPSC, as 4AP (blue) application results in a partial restoration of oEPSC, but not of oIPSC (red lines represent light stimuli, 1ms; scalebar: 100ms, 100 pA).

pathways of HVC. Through this research, we provide the first long-range cellular resolution synaptic connectivity map in the avian DVR, a fundamental step in the effort to identify common themes and differences in synaptic connectivity between the DVR and mammalian neocortical circuits.

## Results

To conduct opsin-assisted circuit mapping in zebra finches, we expressed eGtACR1 in NIf, Uva, mMAN, or Av using AAVs (*Figure 1C*; a cocktail of AAV2.9-Cbh-FLP and AAV2.9-Cbh-fDIO-eGtACR1 injected in HVC afferent areas). 4–6 weeks later, we injected retrograde tracers in HVC projection targets (RA, Area X, and Av) to label HVC-PN classes in different fluorescent channels. After 2–7 days, we obtained acute brain slices and performed whole-cell patch-clamp recordings from visually identified HVC-PNs to examine optically evoked post-synaptic currents (oPSCs). We confirmed that oPSCs measured while holding the membrane voltage at –70 mV were excitatory (oEPSCs) and were mediated by AMPA receptors (current suppressed by bath application of DNQX; *Figure 1C*). Holding the membrane potential to +10 mV allowed us to measure inhibitory currents (oIPSCs) mediated by GABAa receptors (current suppressed by bath application of gabazine; *Figure 1C*).

For each pathway, we report the likelihood of observing oEPSCs and oIPSCs, the amplitude of oEPSCs and oIPSCs, and the excitatory-inhibitory ratio (oEPSC/oIPSC). Although we calculated the paired-pulse ratio, the slow kinetics of eGtACR1 make interpretation of these results difficult; therefore, it is not being reported. We also report whether observed synaptic currents have a monosynaptic component by applying the voltage-gated Na+ channel blocker tetrodotoxin (TTX) followed by the K+ channel blocker 4-aminopyridine (4AP) (*Petreanu et al., 2009*; *Linders et al., 2022*; *Figure 1C*). TTX blocks action potentials and the addition of 4AP blocks the rapid K+-dependent repolarization of the axon. This allows local opsin-driven depolarizations to reach threshold for calcium-dependent vesicle docking and release, thus revealing optically evoked monosynaptic currents (*Linders et al., 2022*). We consistently observed that, while oEPSCs registered in ACSF displayed multiple peaks consistent with a polysynaptic source, currents suppressed by TTX that returned following bath application of 4AP had a monotonic rising slope and a single peak, consistent with a monosynaptic origin of the current (*Figure 1C*).

### Uva monosynaptically innervates HVC_RA neurons

Nucleus Uvaeformis (Uva) is a small polysensory nucleus in the caudal thalamus that is reported to be necessary for song production. Uva is a recipient of somatosensory information from the trigeminal system, visual input from the optic tectum, auditory information from the lateral lemniscus, cholinergic input from the medial habenula, and information about respiratory timing from the ventral respiratory column (*Coleman et al., 2007*; *Faunes and Wild, 2017*; *Wild and Gaede, 2016*; *Wild et al., 2010*; *Wild, 1994*; *Akutagawa and Konishi, 2005*). Its inputs to HVC are thought to be pivotal in song motor control (*Moll et al., 2023*; *Danish et al., 2017*; *Coleman and Vu, 2005*). However, how Uva synaptically influences HVC activity is still poorly understood. We found that Uva neurons are robustly transduced by our AAV expressing eGtACR1 (*Figure 2A*). Thalamic regions around Uva do not project to HVC, allowing us to selectively map Uva's input to the three classes of HVC-PNs.

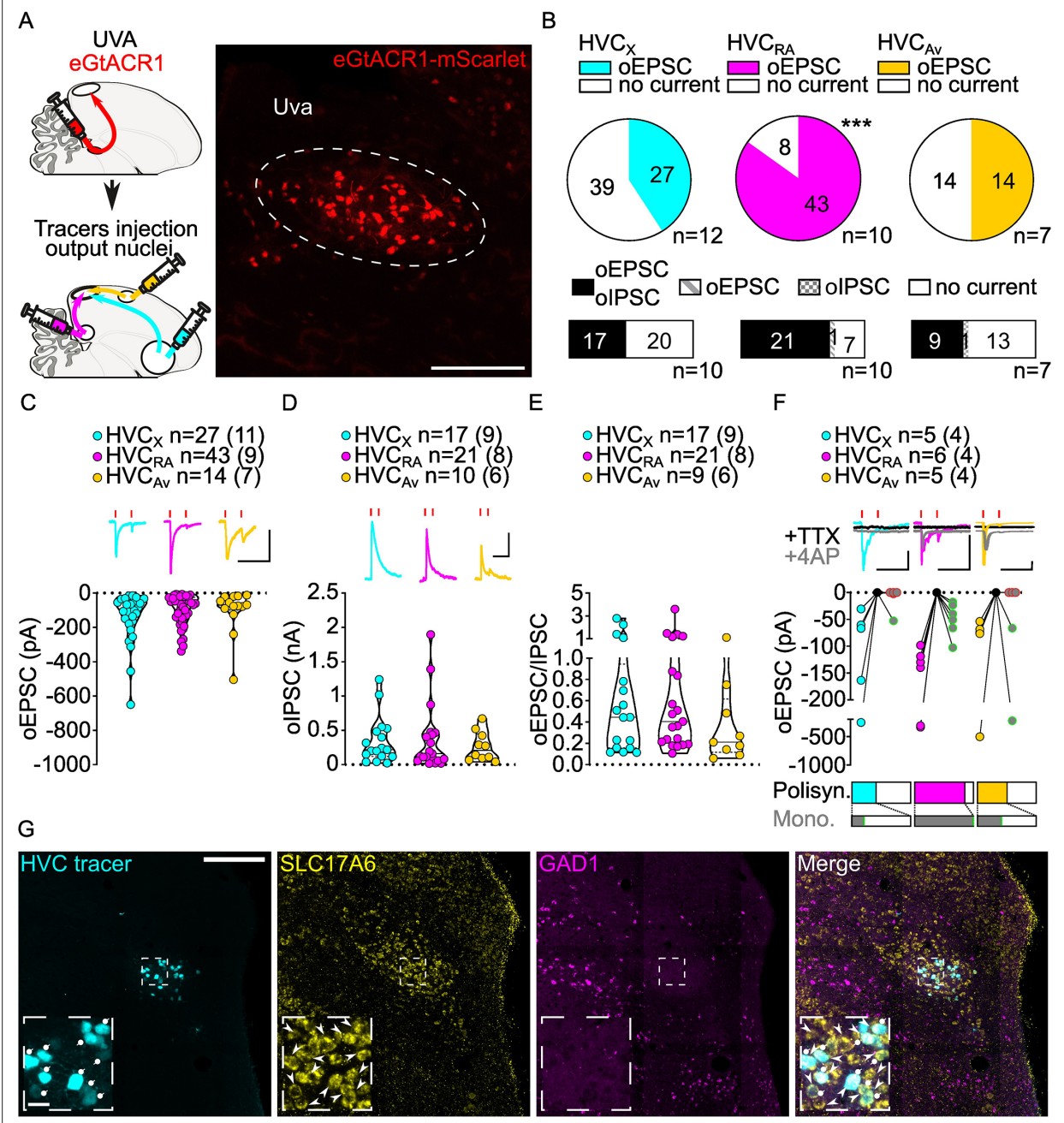

**Figure 2.** Uva afferents reliably elicit polysynaptic oIPSCs and oEPSCs in all three HVC-PN classes and have monosynaptic inputs onto HVC$_{RA}$ neurons. (**A**) (left) Schematic of the experimental timeline, illustrating injection of AAV-eGtACR1 in Uva, followed by retrograde tracer injections in HVC efferent areas and whole-cell patch-clamp recording in acute brain slices; (right) sample image of eGtACR1-mScarlet expression in Uva (scalebar 200 μm). (**B**) (top) pie charts representing the likelihood of observing oEPSCs in HVC$_X$ (cyan), HVC$_{RA}$ (magenta), or HVC$_{Av}$ (yellow) (Fisher's exact test, p<0.001). Numbers in the pie fragments represent the number of cells in which current (colored) or no current (white) was found. Numbers next to the pie charts represent the number of animals from which the data is obtained. (bottom) bar chart representing the number of cells where both oEPSCs and oIPSCs could be elicited (black), only oEPSCs but no oIPSCs (grey lines), only oIPSCs but no oEPSCs (grey checkers), or neither (white), for subsets of cells from the HVC-PN classes' pie charts aligned above (Fisher's exact test, p=0.0244). (**C**) Violin and scatter plot and sample traces reporting average measured oEPSC amplitude of each cell, by cell class (Kruskal-Wallis test H(2)=2.241, p=0.3262; n=cells (animals); red lines represent light stimuli, 1ms; scalebars, 100ms, 100 pA). (**D**) Violin and scatter plot and sample traces of average measured oIPSC amplitude of each cell, by cell class (H(2)=0.5946, p=0.7428; scalebars, 100ms, 100 pA). (**E**) Violin and scatter plot of the ratio of oEPSC and oIPSC peak amplitude of each cell where both are measured and ≠0, per cell class (H(2)=0.2716, p=0.2572). (**F**) (top) sample traces and plot representing the amplitude of post-synaptic currents evoked by lightly-driven release of neurotransmitter from Uva axonal terminals in HVC; oEPSCs amplitudes are reported before (HVC$_X$ cyan, HVC$_{RA}$ magenta, HVC$_{Av}$ yellow) and

*Figure 2 continued on next page*

*Figure 2 continued*

after bath application of TTX (black) and 4AP (grey, green outline indicates polysynaptic oEPSC, see methods), (n=cells (animals); red lines represent light stimuli, 1ms; scalebars, 100ms, 100 pA) (bottom) bar charts representing the likelihood of observing polysynaptic oEPSCs in $HVC_X$ (cyan), $HVC_{RA}$ (magenta), or $HVC_{Av}$ (yellow) (data from panel B), and (grey) likelihood of a subset of the corresponding oEPSCs to be monosynaptic. (**G**) Sample images reporting retrogradely labeled HVC-projecting neurons in UVA (cyan, white circles) together with in situ labeling of glutamatergic (SLC17A6, yellow, white arrowheads) and GABAergic (GAD1, magenta, gray arrowheads) markers transcripts (scalebar 200 µm, inset 20 µm).

The online version of this article includes the following figure supplement(s) for figure 2:

**Figure supplement 1.** oPSCs amplitude and rise latency in each $HVC_{PN}$ class upon optogenetic stimulation of UVA afferents.

In acute brain slices we observed glutamatergic oEPSCs in all three classes of retrogradely identified HVC-PNs (27/66 $HVC_X$, 43/51 $HVC_{RA}$, 14/28 $HVC_{Av}$ neurons; *Figure 2B*, *Figure 2—figure supplement 1*). Optical stimulation of Uva axons resulted in significantly lower probability of observing oEPSCs in $HVC_X$ and $HVC_{Av}$ than in $HVC_{RA}$ neurons (*Figure 2B*, *Figure 2—figure supplement 1*). When we held the membrane voltage to +10 mV, we also observed relatively delayed GABAergic currents (53.93% probability, *Figure 2—figure supplement 1*), consistent with disynaptic inhibition. Uva axon stimulation elicited oIPSCs mainly in cells where we observed oEPSCs (oIPSC contingent with oEPSC: $HVC_X$, 100.0%, $HVC_{RA}$ 96.6%, $HVC_{Av}$, 95.8%; *Figure 2B*).

Despite the markedly higher probability of eliciting currents in $HVC_{RA}$ neurons, oEPSC and oIPSC amplitudes were comparable across all the HVC-PN classes (*Figure 2C, D*, *Figure 2—figure supplement 1*). The excitation/inhibition ratio revealed that Uva terminal stimulation generally led to higher amplitude oIPSCs than oEPSCs (*Figure 2E*, *Figure 2—figure supplement 1*).

We next examined if HVC-PN classes receive monosynaptic input from Uva. Recent studies have suggested different views for how Uva may influence the propagation of activity in HVC. Uva has been proposed to provide high-frequency (30–60 Hz) synchronous synaptic inputs that facilitate activity propagation across $HVC_{RA}$ premotor neurons (*Hamaguchi et al., 2016*). In another proposal, Uva selectively drives activity in only a fraction, ~16%, of $HVC_{RA}$ neurons which are active during syllable transitions (*Moll et al., 2023*). Consistent with the higher likelihood of polysynaptic excitation in $HVC_{RA}$ neurons, we found that oEPSC returned after application of TTX +4 AP in all $HVC_{RA}$ neurons tested ($HVC_{RA}$: 6/6 neurons) while $HVC_X$ and $HVC_{Av}$ were infrequently found to receive monosynaptic input from Uva ($HVC_X$: 1/5 cells, $HVC_{Av}$: 2/5 cells, *Figure 2F*). Our synaptic circuit mapping provides the first conclusive evidence that Uva makes monosynaptic connections with $HVC_{RA}$ neurons. Given that we observed oEPSCs in 84.3% of recorded $HVC_{RA}$ and found monosynaptic connections from Uva onto 100% of the $HVC_{RA}$ neurons tested, our data further suggests that Uva is unlikely to make monosynaptic connections with only a small percentage of $HVC_{RA}$ neurons.

Previous studies have been equivocal to the excitatory, inhibitory, or neuromodulatory nature of the Uva to HVC circuitry (*Moll et al., 2023*; *Coleman et al., 2007*). We did not observe monosynaptic oIPSCs in any cell class, suggesting that Uva provides only a glutamatergic input to HVC ($HVC_X$: 0/3, $HVC_{RA}$: 0/5, $HVC_{Av}$: 0/4, data not shown). We combined retrograde labeling from HVC with in-situ hybridization labeling on brain slices. This revealed that Uva neurons projecting to HVC ($Uva_{HVC}$) selectively expressed the glutamatergic marker SLC17A6, but not the GABAergic marker GAD1 (*Colquitt et al., 2021*; *Figure 2G*). Interestingly, our labeling did not identify any GABAergic neurons within Uva, indicating that $Uva_{HVC}$ may function as an excitatory relay of diverse polysensory input pathways. Although we cannot exclude the possibility that Uva neurons may also co-release neuropeptides, our results define the excitatory synaptic connectivity between Uva and HVC PNs.

## NIf provides excitatory monosynaptic input to all HVC-PNs

We next examined the synaptic transmission between the nucleus NIf and the three classes of HVC-PNs. NIf is a higher order polysensory DVR nucleus located adjacent to the primary auditory DVR (Field L2 - region that receives thalamic input from the avian homologue of the medial geniculate nucleus). NIf is the recipient of afferents from Uva and multiple areas of the auditory DVR and has an essential role in relaying auditory signals to HVC (*Coleman and Mooney, 2004*). The synaptic connection between NIf and HVC is obligatory in a young male zebra finch's ability to form a memory of their father's song (*Zhao et al., 2019*; *Roberts et al., 2012*). This memory is used as birds evaluate their song performances, permitting vocal imitation of song (*Ikeda et al., 2020*). Optogenetic manipulations of NIf inputs to HVC are also sufficient to implant song memories that guide song syllable

imitation (*Zhao et al., 2019*). Although NIf is selectively active during singing, it is not necessary for producing song in adulthood or the song imitation process once the memory of a father's song is acquired in juvenile birds (*Zhao et al., 2019*; *Roberts et al., 2012*; *Mackevicius et al., 2020*; *Otchy et al., 2015*).

To map synaptic connections from NIf to HVC, we expressed eGtACR1 in NIf. While the surrounding area, Field L, has been described to project to the ventral border of HVC (*Vates et al., 1996*; *Fortune and Margoliash, 1995*; *Kelley and Nottebohm, 1979*), we recorded from visually identified, retrogradely labeled HVC projection neurons. Area Avalanche is the only other nearby brain region that also sends projections to HVC, and we carefully checked for a lack of expression in this region in all brain hemispheres used in this study (*Figure 3A*). This provides confidence that the currents we measured are evoked by stimulating neurotransmission from NIf terminals. Light stimulation (1ms) reliably elicited glutamatergic oEPSCs in all three classes of HVC projection neurons (78.6% probability, *Figure 3B*, *Figure 3—figure supplement 1*). We examined oIPSCs in a subset of neurons and found strong GABAergic currents evoked by optical stimulation in all three classes of projection neurons (65.9% probability, *Figure 3B*, *Figure 3—figure supplement 1*). oIPSCs were predominantly found in cells in which we also observed oEPSCs (oIPSC contingent with oEPSC: $HVC_X$ 90.0%; $HVC_{RA}$ 76.5%, $HVC_{Av}$ 94.1%; *Figure 3B*).

We found oEPSC and oIPSC amplitudes to be comparable between $HVC_X$ and $HVC_{Av}$ PNs (*Figure 3C and D*, *Figure 3—figure supplement 1*). However, $HVC_{RA}$ neurons had lower amplitude currents than the other HVC-PN classes (*Figure 3C and D*, *Figure 3—figure supplement 1*). When we computed the amplitude of oEPSCs and oIPSCs for each cell to calculate the excitation/inhibition, we observed that optogenetic stimulation of NIf axonal terminals evoked relatively higher amplitude oIPSCs than oEPSCs in all HVC-PN classes (*Figure 3E*, *Figure 3—figure supplement 1*).

We next examined monosynaptic connectivity from NIf to each HVC-PN. We found that application of TTX followed by 4AP reliably returned oEPSCs in all three HVC projection subtypes (monosynaptic oEPSCs: $HVC_X$: 5/5, $HVC_{RA}$: 4/5, $HVC_{Av}$: 5/5, *Figure 3F*). This widespread monosynaptic connectivity is in stark contrast to thalamic excitatory inputs from Uva that preferentially target $HVC_{RA}$ neurons. Similar to Uva projections, we did not observe monosynaptic oIPSCs ($HVC_X$: 0/5, $HVC_{RA}$: 0/2, $HVC_{Av}$: 0/3, data not shown), suggestive of a glutamatergic projection from NIf to HVC. We combined retrograde labeling from HVC with in situ hybridization labeling on brain slices. NIf neurons projecting to HVC ($NIf_{HVC}$) selectively expressed the glutamatergic marker SLC17A6, but not the GABAergic marker GAD1 (*Colquitt et al., 2021*; *Figure 3G*). This finding is consistent with previous studies indicating that NIf projections result in disynaptic inhibition onto HVC projection neurons (*Coleman and Mooney, 2004*).

Together, these results indicate that intratelencephalic DVR circuits are excitatory and that NIf provides excitatory glutamatergic monosynaptic projections onto all three known classes of HVC-PNs. Given that monosynaptic connectivity across regions of the DVR has not previously been examined, this suggests the possibility that, in contrast to the thalamo-DVR pathways, projections between regions of the DVR might not have cell type selectivity.

## Novel connection from mMAN to Av and selective monosynaptic connections with HVC

To better define the specificity of regional DVR projections, we next mapped synaptic connectivity of two other HVC afferents, mMAN and Av. The DVR is comprised of three large pallial fields separated by distinct lamina: the nidopallium, mesopallium, and the arcopallium. HVC, NIf, and mMAN reside in the nidopallium, and Av is in the mesopallium. HVC and NIf are in caudal regions of the DVR's nidopallium, while mMAN is located at anterior portions of the DVR, some 4–5 mm from HVC. Av, on the other hand, is only ~1.5 mm from HVC but is in a distinct pallial field.

Cell-type-specific synaptic connectivity between nidopallial regions and between mesopallial and nidopallial regions is poorly understood; however, the connectivity of mMAN and Av is better described than other portions of these pallial fields. mMAN receives synaptic input from the thalamic nucleus DMP and is part of a vocal motor DVR-thalamo-DVR loop (mMAN >HVC > RA>DMP > mMAN), a circuit architecture similar to premotor and motor control-related cortico-thalamo-cortical loops in mammals. Although the synaptic connectivity between mMAN and HVC has not been well studied, lesions of mMAN in juvenile zebra finches result in song imitation deficits and lesions in adult

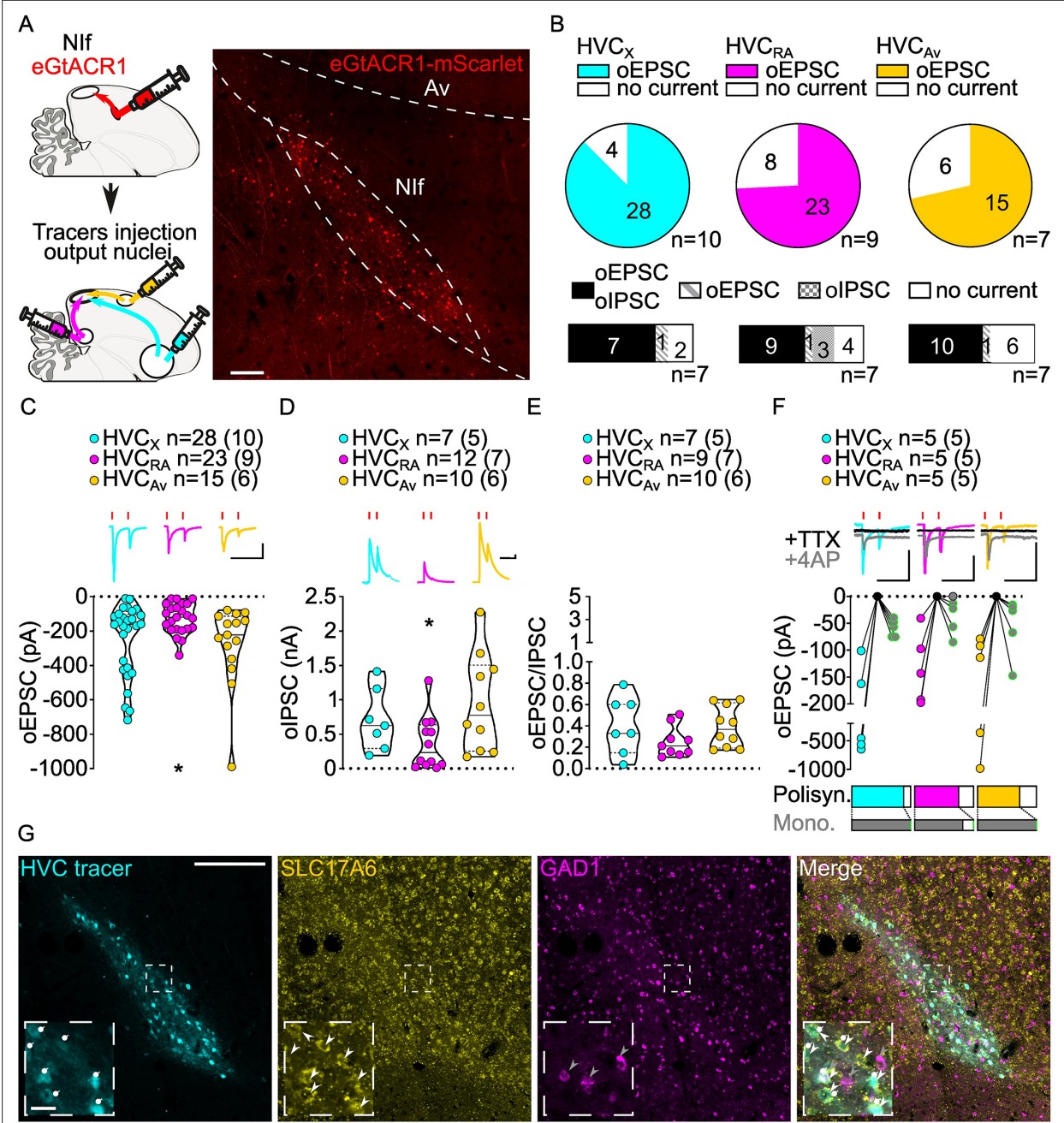

**Figure 3.** Stimulation of NIf inputs to HVC reliably elicits polysynaptic oIPSCs and oEPSCs, and monosynaptic oEPSCs, in all three HVC-PN classes. (**A**) (left) Schematic of the experimental timeline, illustrating injection of AAV-eGtACR1 in NIf, followed by retrograde tracer injections in HVC efferent areas and whole-cell patch-clamp recording in acute brain slices; (right) sample image of eGtACR1-mScarlet expression in NIf (scalebar 200 µm). (**B**) (top) pie charts representing the likelihood of observing oEPSCs in HVC$_X$ (cyan), HVC$_{RA}$ (magenta) or HVC$_{Av}$ (yellow) (Fisher's exact test, p=0.2783). Numbers in the pie fragments represent the number of cells in which current (colored) or no current (white) was found. Numbers next to the pie charts represent the number of animals from which the data is obtained. (bottom) bar chart representing the number of cells where both oEPSCs and oIPSCs could be elicited (black), only oEPSCs but no oIPSCs (grey lines), only oIPSCs but no oEPSCs (grey checkers), or neither (white), for subsets of cells from the HVC-PN classes' pie charts aligned above (Fisher's exact test, p=0.5841). (**C**) Violin and scatter plot and sample traces reporting average measured oEPSC amplitude of each cell, by cell class (Kruskal-Wallis test, H(2)=6.135, p=0.0465; n=cells (animals); red lines represent light stimuli, 1ms; scalebars, 100ms, 100 pA). (**D**) Violin and scatter plot and sample traces of average measured oIPSC amplitude of each cell, by cell class (H(2)=6.182, p=0.0455; scalebars, 100ms, 100 pA). (**E**) Violin and scatter plot of the ratio of oEPSC and oIPSC peak amplitude of each cell where both are measured and ≠0, per cell class (H(2)=3.305, p=0.1916). (**F**) (top) sample traces and plot representing the amplitude of post-synaptic currents evoked by lightly-driven release of neurotransmitter from NIf axonal terminals in HVC; oEPSCs amplitudes are reported before (HVC$_X$ cyan, HVC$_{RA}$ magenta, HVC$_{Av}$ yellow) and

*Figure 3 continued*

after bath application of TTX (black) and 4AP (grey, green outline indicates polysynaptic oEPSC, see Materials and methods), (n=cells (animals); blue lines represent light stimuli, 1ms; scalebbars, 100ms, 100 pA) (bottom) bar charts representing the likelihood of observing polysynaptic oEPSCs in HVC$_X$ (cyan), HVC$_{RA}$ (magenta) or HVC$_{Av}$ (yellow) (data from panel B), and (grey) likelihood of a subset of the corresponding oEPSCs to be monosynaptic. (**G**) Sample images reporting retrogradely labeled HVC-projecting neurons in NIf (cyan, white circles) together with in situ labeling of glutamatergic (SLC17A6, yellow, white arrowheads) and GABAergic (GAD1, magenta, gray arrowheads) markers transcripts (scale bar 200 μm, inset 20 μm).

The online version of this article includes the following figure supplement(s) for figure 3:

**Figure supplement 1.** oPSCs amplitude and rise latency in each HVC$_{PN}$ class upon optogenetic stimulation of NIf afferents.

Bengalese finches increase song syntax variability, suggesting a role in selection and planning of vocal motor sequencing (*Foster and Bottjer, 2001*; *Koparkar et al., 2024*). Av is one of the more recently described song-related regions, and unlike Uva, NIf, and mMAN, it is reciprocally connected to HVC. The role of Av is still poorly explored, but it is understood to be embedded in a higher order auditory portion of the DVR and thought to be an important node in the circuit comparing efferent copies of motor commands from HVC to auditory feedback (*Bauer et al., 2008*; *Akutagawa and Konishi, 2010*).

While evaluating connectivity between HVC, mMAN, and Av, we discovered that mMAN also provides a strong projection to Av (*Figure 4*). Approximately half of the identified projection neurons in mMAN project to Av, and our tracing experiments reveal that there are only partially overlapping classes of projection neurons in mMAN: mMAN$_{HVC}$ PNs, mMAN$_{Av}$ PNs, and mMAN$_{HVC/AV}$ PNs. This newly identified pathway appears positioned to provide the auditory system with information about song motor commands via a DVR-thalamo-DVR loop (RA >DMP > mMAN>Av). It also opens the potential for synaptic loops relaying through HVC between these regions.

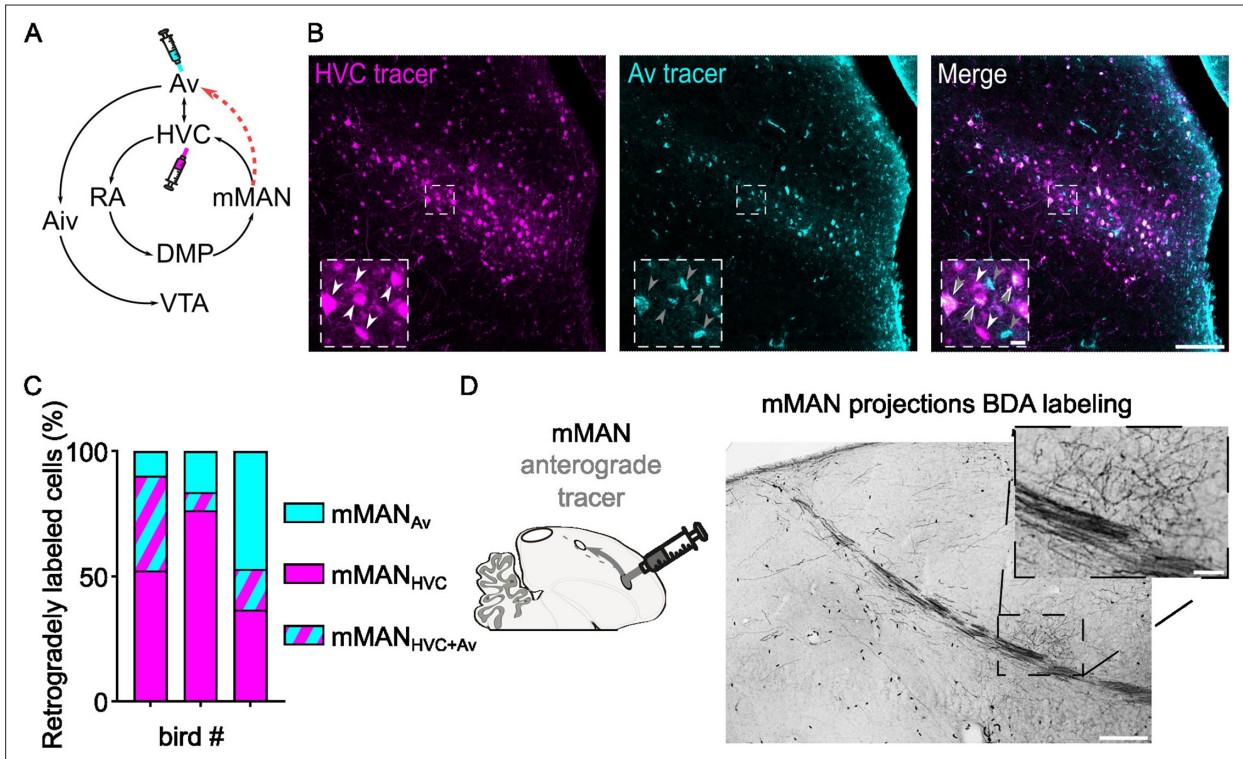

**Figure 4.** Identification of mMAN neuronal subpopulations projecting to Av and to HVC. (**A**) Schematic of the known mMAN afferent and efferent circuitry, together with the proposed new connection towards Av (red dashed arrow), and schematized injection of retrograde tracers in HVC and Av. (**B**) Sample images of retrogradely labeled mMAN cells projecting to HVC (magenta) or to Av (cyan), and merged image. Insets report magnified selection (dashed square box in the images) illustrating the potential three subpopulations identified: mMAN$_{HVC}$ (white arrowheads), mMAN$_{Av}$ (grey arrowheads) and mMAN$_{HVC+Av}$ (overimposed white and grey arrowheads) (scalebars: 100 μm, inset: 10 μm). (**C**) Bar chart displaying the number of retrogradely mMAN$_{HVC}$, mMAN$_{Av}$, and mMAN$_{HVC+Av}$ labeled cells, in three birds (averaged across hemispheres, 2–6 slices/bird). (**D**) BDA labeling of anterograde projection to Av from mMAN, inset magnifies the terminal field in Av (scale bars: 200 μm, inset: 50 μm).

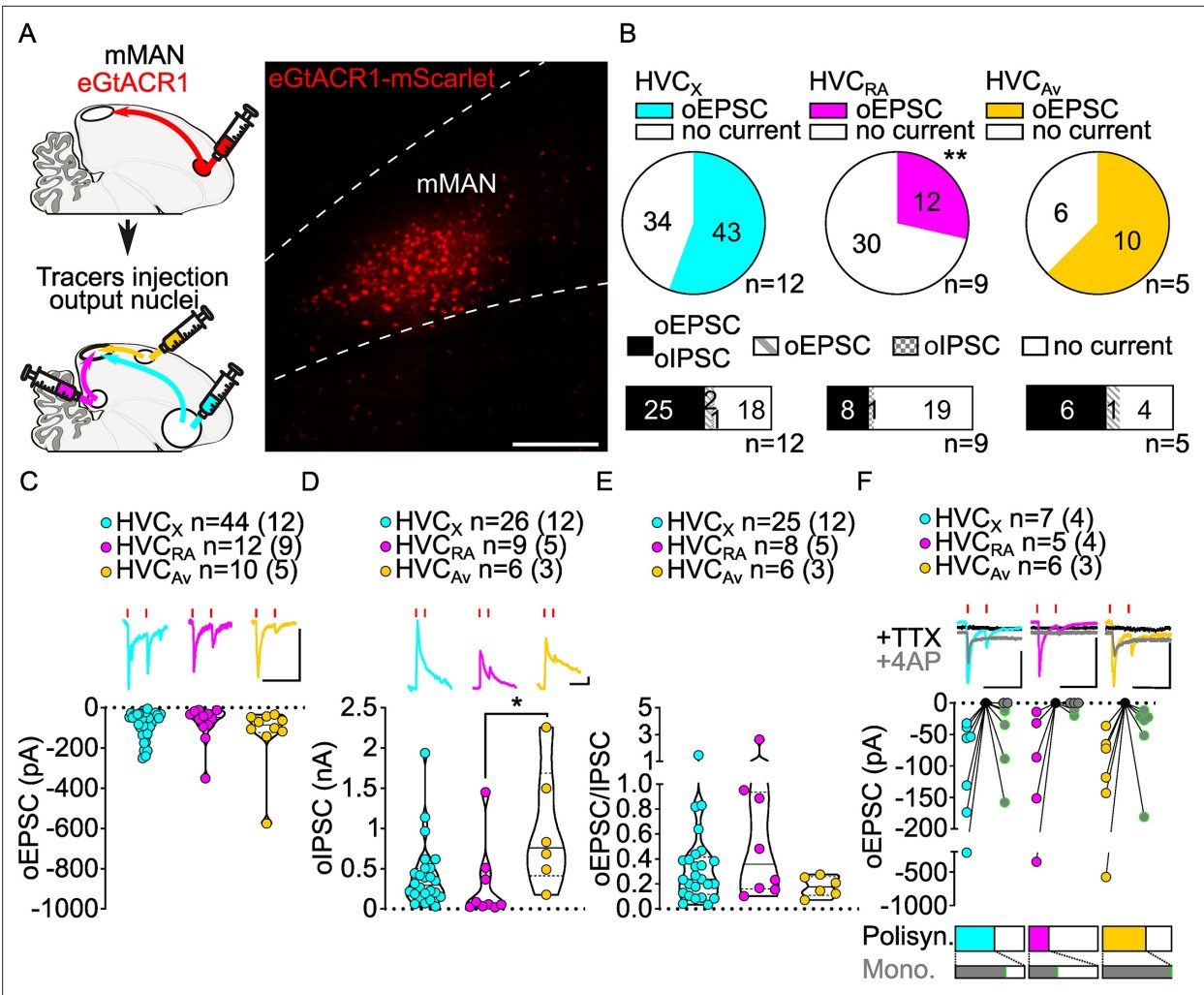

**Figure 5.** mMAN inputs elicit polysynaptic oIPSCs and oEPSCs in all three HVC-PN classes and monosynaptically excite HVC$_X$ and HVC$_{Av}$ neurons. (**A**) (left) Schematic of the experimental timeline, illustrating injection of AAV-eGtACR1 in MMAN, followed by retrograde tracer injections in HVC efferent areas and whole-cell patch-clamp recording in acute brain slices; (right) sample image of eGtACR1-mScarlet expression in MMAN (scale bar 200 μm). (**B**) (top) pie charts representing the likelihood of observing oEPSCs in HVC$_X$ (cyan), HVC$_{RA}$ (magenta), or HVC$_{Av}$ (yellow) (Fisher's exact test, p=0.0088). Numbers in the pie fragments represent the number of cells in which current (colored) or no current (white) was found. Numbers next to the pie charts represent the number of animals from which the data is obtained. (bottom) bar chart representing the number of cells where both oEPSCs and oIPSCs could be elicited (black), only oEPSCs but no oIPSCs (grey lines), only oIPSCs but no oEPSCs (grey checkers), or neither (white), for subsets of cells from the HVC-PN classes' pie charts aligned above (Fisher's exact test, p=0.1054). (**C**) Violin and scatter plot and sample traces reporting average measured oEPSC amplitude of each cell, by cell class (Kruskal-Wallis test H(2)=2.659, p=0.2646; n=cells (animals); red lines represent light stimuli, 1ms; scale bars, 100ms, 100 pA). (**D**) Violin and scatter plot and sample traces of average measured oIPSC amplitude of each cell, by cell class (H(2)=8.598, p=0.0136, HVC$_{RA}$ vs. HVC$_{Av}$ p=0.0102; scalebars, 100ms, 100 pA). (**E**) Violin and scatter plot of the ratio of oEPSC and oIPSC peak amplitude of each cell where both are measured and ≠0, per cell class (H(2)=2.443, p=0.2759). (**F**) (top) sample traces and plot representing the amplitude of post-synaptic currents evoked by lightly-driven release of neurotransmitter from MMAN axonal terminals in HVC; oEPSCs amplitudes are reported before (HVC$_X$ cyan, HVC$_{RA}$ magenta, HVC$_{Av}$ yellow) and after bath application of TTX (black) and 4AP (grey, green outline indicates polysynaptic oEPSC, see Materials and methods), (n=cells (animals); blue lines represent light stimuli, 1ms; scalebars, 100ms, 100 pA) (bottom) bar charts representing the likelihood of observing polysynaptic oEPSCs in HVC$_X$ (cyan), HVC$_{RA}$ (magenta) or HVC$_{Av}$ (yellow) (data from panel B), and (grey) likelihood of a subset of the corresponding oEPSCs to be monosynaptic.

The online version of this article includes the following figure supplement(s) for figure 5:

**Figure supplement 1.** oPSCs amplitude and rise latency in each HVC$_{PN}$ class upon optogenetic stimulation of mMAN afferents.

Brain regions nearby mMAN do not send projections to HVC, and we were able to achieve robust expression of eGtACR1 in mMAN (*Figure 5A*), allowing us to selectively examine the synaptic connectivity between mMAN and HVC-PNs. Optical stimulation of mMAN axon terminals in brain slices revealed glutamatergic oEPSCs in all retrogradely identified projection neuron classes, albeit with significantly lower success rate compared to what we observed when stimulating NIf terminals (48.15% probability; mMAN vs. NIf oEPSC probability: Fisher's exact test p<0.001). In addition, we found that oEPSCs were more likely evoked in recordings from HVC$_X$ and HVC$_{Av}$ than from HVC$_{RA}$ neurons (*Figure 5B*, *Figure 5—figure supplement 1*).

Optogenetic stimulation of mMAN axon terminals also revealed GABAergic currents in all three HVC-PN classes (48.24% probability). Similar to NIf and Uva inputs, oIPSCs elicited by mMAN axons were most often found in cells where oEPSCs were also observed (oIPSC contingent with oEPSC: HVC$_X$: 93.5%, HVC$_{RA}$: 96.4%, HVC$_{Av}$: 90.9%; *Figure 5B*).

While oEPSCs amplitudes across HVC-PN classes were statistically indistinguishable (*Figure 5C*), we found that oIPSCs in HVC$_{Av}$ had significantly higher amplitude than those evoked in HVC$_{RA}$ neurons (*Figure 5D*, *Figure 5—figure supplement 1*). Despite this difference, when computing the E/I ratio, we did not see any significant difference across HVC-PNs, and like the other inputs examined, GABAergic currents had higher amplitude than glutamatergic currents (*Figure 5E*, *Figure 5—figure supplement 1*).

We next asked whether mMAN monosynaptically contacted any class of HVC-PNs. We found evidence that mMAN monosynaptically contacts all three classes of HVC PNs. The most reliable monosynaptic transmission is between mMAN and HVC$_{Av}$ neurons and HVC$_X$ neurons (HVC$_{RA}$: 2/5, HVC$_X$: 5/7, HVC$_{Av}$: 6/6, *Figure 5F*). We did not observe monosynaptic oIPSCs (HVC$_X$: 0/5, HVC$_{RA}$, 0/5, HVC$_{Av}$: 0/6, data not shown), and confirmed that mMAN neurons projecting to HVC are gluta-matergic using in-situ hybridization combined with retrograde tracing from injection of tracer in HVC (*Figure 5—figure supplement 1*). Together, this data indicates that mMAN predominantly provides robust monosynaptic excitatory input to HVC$_{Av}$ and HVC$_X$ neurons. HVC$_{Av}$ neurons only comprise ~3% of the neurons in HVC (*Roberts et al., 2017*). Therefore, finding that 100% of the HVC$_{Av}$ neurons tested receive monosynaptic input from mMAN, suggesting that this is likely to be a highly selective synaptic target of mMAN inputs to HVC. Given that we have now also identified a projection from mMAN directly to Av, this supports a model in which mMAN is providing Av song-related signals directly and indirectly via projections onto HVC$_{Av}$ neurons.

Lastly, we mapped the synaptic inputs from Av onto HVC-PNs. Avalanche lacks clear anatomical boundaries, and our viral expression did spread into surrounding brain regions, but only brain hemi-spheres in which we could verify a lack of expression in NIf were included in this study (*Figure 6A*). Whole-cell patch-clamp recordings from identified HVC-PN classes in brain slices revealed glutama-tergic oEPSCs in all three cell classes, yet most of the cells were not excited by Av terminals (39.18% probability; Av vs. NIf oEPSC probability: Fisher's exact test p<0.001). However, the probability of evoking oEPSCs was similar among HVC-PN classes (*Figure 6B*, *Figure 6—figure supplement 1*). We tested the presence of oIPSCs in a subset of cells and found GABAergic currents in all three projection subtypes (43.24% probability, *Figure 6—figure supplement 1*). As in other pathways already described, we observed oIPSCs mainly in HVC-PNs also showing oEPSCs (oIPSC contingent with oEPSC: HVC$_X$ 100.0% HVC$_{RA}$ 90.0%, HVC$_{Av}$: 84.6%; *Figure 6B*).

While oEPSCs amplitudes were broadly similar among HVC-PN classes (*Figure 6C*), oIPSCs had significantly smaller amplitude in HVC$_{RA}$ compared to those evoked in HVC$_X$ neurons (*Figure 6D*, *Figure 6—figure supplement 1*). Consistently, this was also reflected in a significantly higher E/I ratio in HVC$_{RA}$ compared to HVC$_X$. Similar to the other afferents, Av axon terminal stimulation elic-ited stronger GABAergic currents than glutamatergic currents across HVC-PN classes (*Figure 6E*, *Figure 6—figure supplement 1*).

When examining which HVC-PN neuron classes received monosynaptic input from Av, we consis-tently observed monosynaptic transmission onto HVC$_X$ neurons (HVC$_X$ 4/5 cells monosynaptically contacted). Monosynaptic transmission onto HVC$_{RA}$ and HVC$_{Av}$ was less common (HVC$_{RA}$ 2/5, HVC$_{Av}$ 2/6; *Figure 6F*). Previous reports indicated that Av neurons projecting to HVC are excitatory (*Roberts et al., 2017*). Consistent with this, we didn't observe any monosynaptic oIPSC across HVC-PNs (monosynaptic oIPSCs: HVC$_X$ 0/4, HVC$_{RA}$ 0/3, HVC$_{Av}$ 0/3, data not shown) and further confirmed that Av$_{HVC}$ neurons are glutamatergic by in situ hybridization labeling of retrogradely identified neurons

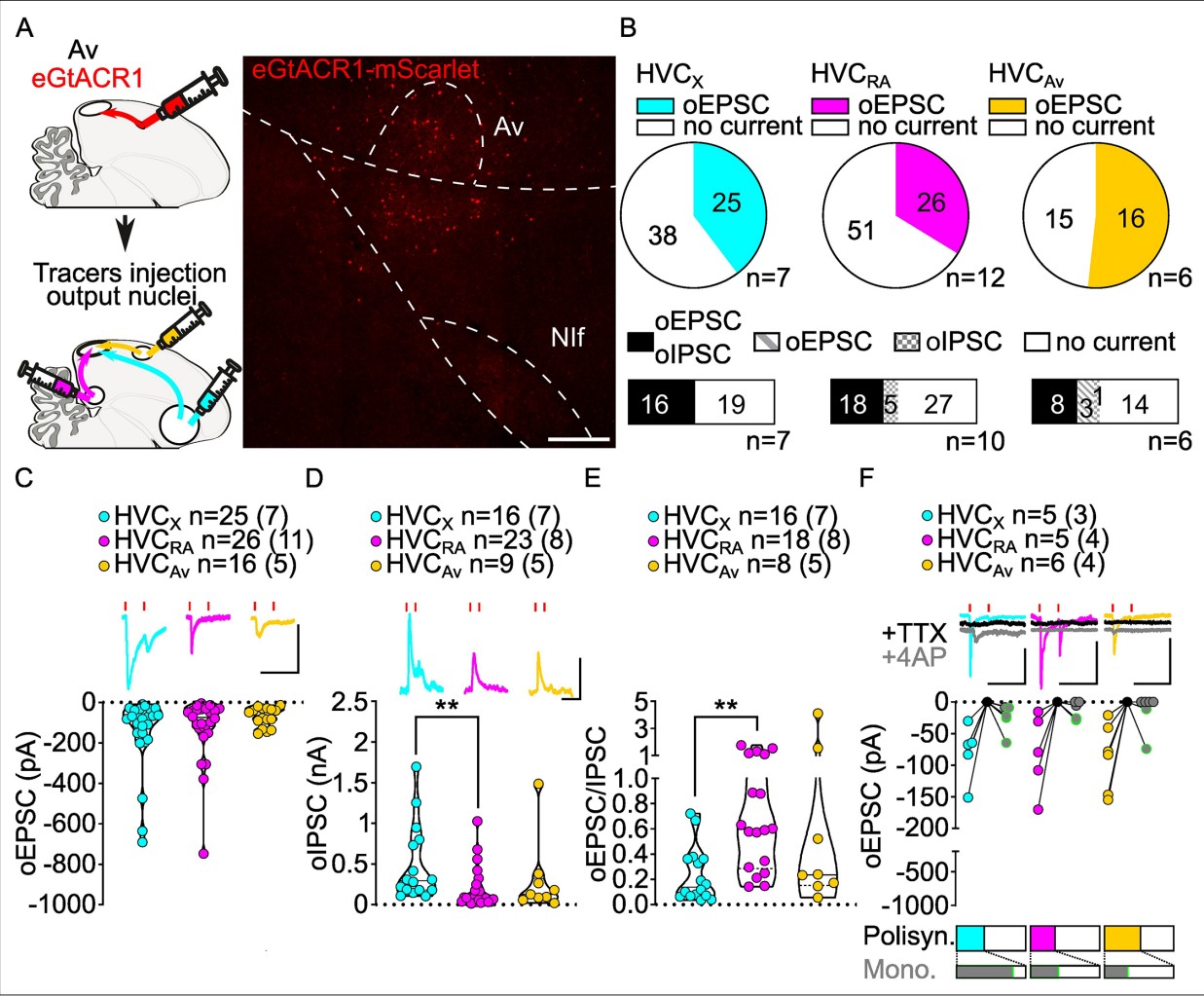

**Figure 6.** Av inputs elicit polysynaptic oIPSCs and oEPSCs in all three HVC-PN classes and monosynaptically excite mainly HVC_X neurons. (**A**) (left) Schematic of the experimental timeline, illustrating injection of AAV-eGtACR1 in Av, followed by retrograde tracer injections in HVC efferent areas and whole-cell patch-clamp recording in acute brain slices; (right) sample image of eGtACR1-mScarlet expression in Av (scalebar 200 µm). (**B**) (top) pie charts representing the likelihood of observing oEPSCs in HVC_X (cyan), HVC_RA (magenta) or HVC_Av (yellow) (Fisher's exact test, p=0.2243). Numbers in the pie fragments represent the number of cells in which current (colored) or no current (white) was found. Numbers next to the pie charts represent the number of animals from which the data is obtained. (bottom) bar chart representing the number of cells where both oEPSCs and oIPSCs could be elicited (black), only oEPSCs but no oIPSCs (grey lines), only oIPSCs but no oEPSCs (grey checkers), or neither (white), for subsets of cells from the HVC-PN classes' pie charts aligned above (Fisher's exact test, p=0.0550). (**C**) Violin and scatter plot and sample traces reporting average measured oEPSC amplitude of each cell, by cell class (Kruskal-Wallis test H(2)=1.103, p=0.5760; n=cells (animals); red lines represent light stimuli, 1ms; scalebars, 100ms, 100 pA). (**D**) Violin and scatter plot and sample traces of average measured oIPSC amplitude of each cell, by cell class (H(2)=10.09, p=0.0064, HVC_X vs. HVC_RA p=0.0047; scalebars, 100ms, 100 pA). (**E**) Violin and scatter plot of the ratio of oEPSC and oIPSC peak amplitude of each cell where both are measured and ≠0, per cell class (H(2)=10.64, p=0.0049, HVC_X vs. HVC_RA p=0.0033). (**F**) (top) sample traces and plot representing the amplitude of post-synaptic currents evoked by lightly-driven release of neurotransmitter from Av axonal terminals in HVC; oEPSCs amplitudes are reported before (HVC_X cyan, HVC_RA magenta, HVC_Av yellow) and after bath application of TTX (black) and 4AP (grey, green outline indicates polysynaptic oEPSC, see Materials and methods), (n=cells (animals); blue lines represent light stimuli, 1ms; scale bars, 100ms, 100 pA) (bottom) bar charts representing the likelihood of observing polysynaptic oEPSCs in HVC_X (cyan), HVC_RA (magenta), or HVC_Av (yellow) (data from panel B), and (grey) likelihood of a subset of the corresponding oEPSCs to be monosynaptic.

The online version of this article includes the following figure supplement(s) for figure 6:

**Figure supplement 1.** oPSCs amplitude and rise latency in each HVC_PN class upon optogenetic stimulation of Av afferents.

(*Figure 6—figure supplement 1*). Our data shows that Av provides monosynaptic excitatory input to HVC$_X$ neurons with high frequency and makes monosynaptic connections with HVC$_{RA}$ and HVC$_{Av}$ neurons less frequently.

Taken together, this research provides a rich dataset mapping the polysynaptic and monosynaptic connectivity of the input and output pathways of HVC – the best studied portions of the avian song circuitry – as well as a newly discovered pathway between DVR circuits that are important for vocal learning (*Figure 7*).

## Discussion

Speech and language are controlled by interconnected neocortical and subcortical networks, including dorsal and ventral cortical pathways for articulation and lexical processing, and pathways looping through the thalamus, basal ganglia, and cerebellum (*Konopka and Roberts, 2016*; *Hickok and Poeppel, 2007*; *Poeppel et al., 2012*; *Hertrich et al., 2020*). The synaptic connectivity between specific populations of neurons among these circuits is going to be difficult to map. Yet, this will ultimately be necessary for understanding the neuroanatomical basis and circuit computations associated with speech and language. Although the avian DVR and mammalian neocortex have distinct developmental origins (ventral and dorsal pallium, respectively), HVC PNs neurons have similar gene expression and connectivity patterns to the intratelencephalic neurons described in layers 2–6 of the mammalian neocortex (*Colquitt et al., 2021*), suggesting evolutionary convergence in circuit computations needed for complex behaviors like vocal imitation. Thus, there is much to be gained by mapping the connectivity between the specific populations of neurons known to be essential in the sensory and sensorimotor process of song imitation.

Using large-scale, multi-pathway, and long-range functional synaptic circuit mapping, we report cell-type-specific synaptic connectivity across forebrain sensorimotor networks necessary for learning and producing learned vocalizations. The songbird motor nucleus HVC is crucial for song production. Its partial or complete lesion results in song disruption and permanent loss, respectively (*Aronov et al., 2008*; *Simpson and Vicario, 1990*; *Basista et al., 2014*). Electrical stimulation of HVC can halt song (*Vu et al., 1994*; *Ashmore et al., 2005*; *Vu et al., 1998*). Further, HVC is essential in juvenile song learning (*Roberts et al., 2017*; *Garcia-Oscos et al., 2021*; *Roberts et al., 2012*; *Sánchez-Valpuesta et al., 2019*; *Tanaka et al., 2018*). We show that long-range sensory pathways to HVC are excitatory and contact HVC intratelencephalic PNs. We find that these connections are not stochastic but are instead strongly biased to make synapses with premotor (HVC$_{RA}$), auditory (HVC$_{AV}$), or basal ganglia (HVC$_X$) projection pathways. Lastly, we find a remarkable correspondence in the probability of finding oEPSCs and oIPSCs in the same postsynaptic neurons, indicative of highly interconnected cell assemblies at postsynaptic targets of long-range connections within the DVR.

In addition to finding widespread polysynaptic transmission upon stimulation of any of the afferents, we endeavored to reveal the wiring diagram of HVC inputs by isolating the monosynaptic components of these currents (*Figure 7*). We found surprisingly specific and compartmentalized monosynaptic neurotransmission between each input and HVC-PN classes. NIf terminals monosynaptically contacted all three HVC-PN classes. Uva predominantly makes monosynaptic connections with HVC$_{RA}$ neurons, while mMAN most frequently makes monosynaptic connections with HVC$_{Av}$ and HVC$_X$ PNs but appears to only sparsely project to HVC$_{RA}$ neurons. Lastly, Av predominantly makes monosynaptic connections with HVC$_X$ neurons and connects less frequently with HVC$_{RA}$ and HVC$_{Av}$ neurons. When combined with our discovery of a new pathway from mMAN to Av, our data draws a scenario of multiple intermingled loops coexisting across HVC input and output circuits.

Thalamic inputs from Uva may directly affect the song motor pathway through monosynaptic projections onto HVC$_{RA}$ neurons. The potential relevance of this specific connection has been recently studied (*Moll et al., 2023*; *Hamaguchi et al., 2016*). Additionally, the seldom yet significant direct inputs to HVC$_{Av}$ neurons we identify are interesting considering the proposed role of HVC$_{Av}$ neurons in providing a forward model of song to the auditory system (*Roberts et al., 2017*). Uva is also known to project to both NIf and Av. Uva's monosynaptic connections to classes of HVC PNs may therefore directly integrate with the propagation of forward models of song timing to the auditory system, particularly during the learning of syllable transitions (*Roberts et al., 2017*).

mMAN has recently been found to contribute to the ordering of song syllables (*Koparkar et al., 2024*). This could be attributable to the strong connections identified here between mMAN and

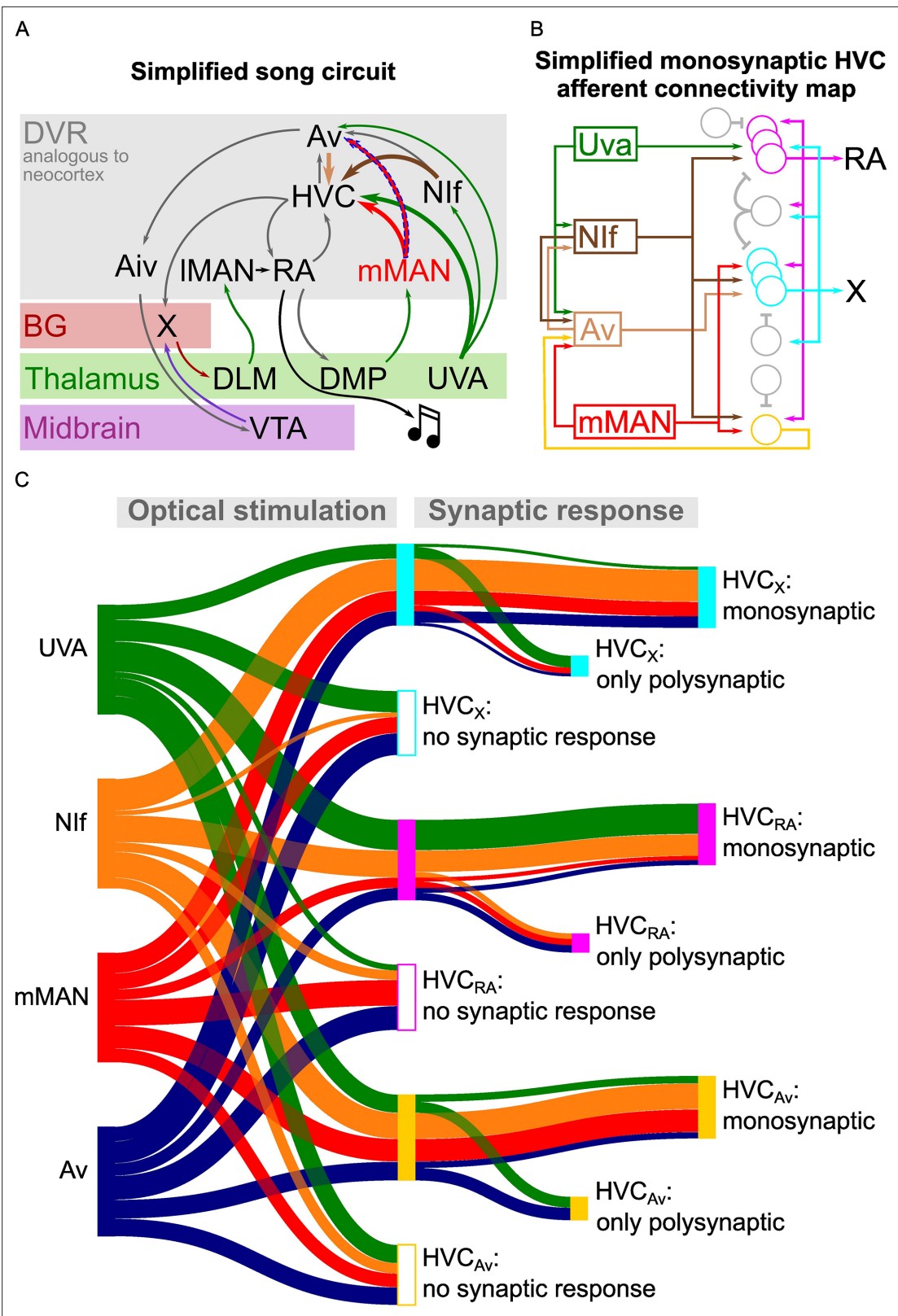

**Figure 7.** Song circuitry summary and monosynaptic connectivity map of afferents to HVC-PNs. (**A**) Schematic representing a simplified version of the long-range connectivity map for song. Colors represent the anatomical location of each projection: gray: DVR, red: basal ganglia, green: thalamus, purple: midbrain. HVC afferents described in this manuscript are highlighted by large arrows. The novel mMAN-Av projection is highlighted by a dashed outline. (**B**) Schematic of the HVC afferent connectivity map resulting from the present work, complemented with projections between HVC afferent

*Figure 7 continued on next page*

*Figure 7 continued*

areas (on the left) and between HVC projection neurons as per previous reports (on the right). For conceptualization purposes, afferent connectivity to HVC-PNs is shown only when the rate of monosynaptic connectivity reaches 50% of neurons examined: NIf monosynaptically contacts all three HVC-PNs, while Uva is preferentially monosynaptically connected to $HVC_{RA}$, mMAN to $HVC_X$ and $HVC_{Av}$, and Av to $HVC_X$. (**C**) Sankey diagram displaying the prevalence of connectivity for each input and cell subtype combination, based on polysynaptic and monosynaptic connectivity rates described in *Figures 2–3 and 5–6*.

The online version of this article includes the following figure supplement(s) for figure 7:

**Figure supplement 1.** Cross-comparison of afferent polysynaptic connectivity within cell types (data from *Figures 2, 3, 5 and 6*).

---

$HVC_{Av}$ PNs and from mMAN directly to Av. $HVC_{Av}$ neurons are important for learning the timing of elements within song. Lesions of $HVC_{Av}$ neurons in juvenile birds disrupt learning of syllable syntax yet do not have a strong effect on learning the spectral features of syllables (*Roberts et al., 2017*). Further, lesions block deafening-induced degradation of songs' temporal features and recovery of song timing following song-contingent feedback perturbations (*Roberts et al., 2017*). Thus, our analysis of synaptic connectivity provides important anatomical connections for a circuit critical for temporal ordering of vocalizations during learning and in adulthood.

mMAN and Av's preferential connection to $HVC_X$ PNs may create a bridge linking the thalamus, the auditory system, and the basal ganglia circuits important for song plasticity. mMAN receives projections from the nucleus DMP of the thalamus, which itself receives projections from RA. mMAN is therefore positioned to receive a copy of descending vocal motor commands transmitted through DMP, while Av may relay integrated information about auditory feedback and their correspondence with motor commands for song that it receives from mMAN and from $HVC_{Av}$ neurons. Our findings place mMAN and Av in an ideal position to relay those signals to Area X, potentially contributing to song plasticity.

In the data shown in *Figures 2, 3 and 5–6* we examined the probability and strength of oPSCs within each HVC-PNs class. We found that projection neurons in NIf, Uva, mMAN and Av consistently establish polysynaptic contact with all three HVC-PN classes. We reliably observed both glutamatergic and GABAergic polysynaptic neurotransmission upon afferent axonal terminal stimulation. Organizing our data by HVC PN types (*Figure 7—figure supplement 1*) reveals a bias in afferent inputs onto these different neuronal pathways. We find that $HVC_X$ and $HVC_{AV}$ neurons are most frequently excited by NIf afferents, while $HVC_{RA}$ are most frequently excited by Uva, and unlikely to be excited by mMAN terminal stimulation. $HVC_X$ and $HVC_{Av}$ neurons are most strongly driven by NIf, while $HVC_{RA}$ neurons receive relatively uniform amplitude oEPSCs from all four input pathways examined. Lastly, stimulation of NIf and mMAN terminals resulted in higher amplitude inhibition onto $HVC_{Av}$ neurons than upon Uva or Av terminals stimulation (*Figure 7—figure supplement 1*).

This data points towards the existence of a rich interneuronal inhibitory network that is recruited by stimulating HVC's afferents, evoking GABAergic transmission onto all HVC-PN classes. This is in line with previous reports that found HVC-PNs activity to be tightly regulated by interneurons (*Kosche et al., 2015*; *Vallentin et al., 2016*; *Mooney and Prather, 2005*). Interestingly, across all inputs studied, the likelihood of finding oIPSCs was tightly dependent on finding oEPSCs in the same cell. Previous studies have indicated that HVC PNs are disynaptically inhibited by other HVC-PNs, a feature of synaptic connectivity that is likely fundamental to the propagation of excitation in the network during song production and in the song learning process (*Kosche et al., 2015*; *Mooney and Prather, 2005*). When afferents cause excitation of HVC PNs this would therefore cause an immediate rise in the levels of disynaptic inhibition in the surrounding HVC PNs. Similar feedback inhibition dynamics have been extensively reported in cortical circuits, where this mechanism serves a crucial role of controlling the excitatory-inhibitory balance of local principal neurons (*Tremblay et al., 2016*; *Berger et al., 2010*; *Kapfer et al., 2007*; *Silberberg and Markram, 2007*). Afferents that excite HVC PNs may also contact neighboring interneurons. A similar feedforward inhibition has also been reported in the cortex, where both pyramidal neurons and interneurons projecting to them are recruited by long-range pathways (*Anastasiades et al., 2018*; *Delevich et al., 2015*). While feedback inhibition is poised to monitor and modulate the output of local principal neurons, feedforward inhibition is proposed to work as a coincidence detector improving the sensitivity of the network by filtering asynchronous inputs (*Tremblay et al., 2016*; *Bruno and Sakmann, 2006*; *Bruno and Simons, 2002*; *Cardin et al., 2010*; *Pinto et al., 2000*; *Pinto et al., 2003*).

Distinguishing among these potential feedback and feedforward circuit motifs, as well as fully characterizing the role of afferents for HVC local circuit dynamics, will require monosynaptic connectivity mapping of afferents onto HVC interneurons. The ability to do this systematically in HVC is limited by the lack of genetic methods for labeling interneurons (*Dimidschstein et al., 2016*). *Colquitt et al., 2021* have recently identified multiple GABAergic cell subtypes in HVC, and a molecular handle is needed to experimentally approach the study of their synaptic connectivity. Nonetheless, our data suggests the existence of compartmentalized neuronal ensembles including both PNs and interneurons projecting to those PNs.

The synaptic organization of excitatory-inhibitory connections in HVC seems to allow the integration of afferent inputs within hyper-local networks or modules. This type of connectivity may make the circuit more robust to perturbations (*Stauffer et al., 2012*). A potential caveat is that we performed all our recordings in 230-μm-thick sagittal brain slices, and it is thought that HVC-PNs exhibit sequential activity that is organized in the rostrocaudal axis, while interneurons are speculated to be coordinated in the medio-lateral axis (*Day et al., 2013*). Our resection of this mediolateral organization may account for the lack of lateral inhibition found in our data. For example, our optogenetic stimulation elicits oIPSCs mostly in the same neurons where oEPSCs are observed, but not in the others, which may have received inhibition from interneurons sitting in adjacent brain slices. Lateral inhibition is normally observed in circuits displaying feedback inhibition, and previous reports indicate that HVC-PNs disynaptically inhibit other HVC-PNs (*Kosche et al., 2015*; *Mooney and Prather, 2005*). It is, however, possible that differences between neocortical and HVC microcircuitry may reduce the significance of lateral inhibition, explaining why we only seldomly observed it.

Our measurements of oPSCs onset delays from the light stimulation offer some insight into the possible wiring of afferents and intra-HVC connectivity. $HVC_{RA}$ neurons displayed the fastest oEPSCs latency upon Uva axon terminal stimulation, consistent with their preferential monosynaptic connectivity. Likewise, $HVC_X$ neurons display the fastest latency upon Nlf axon terminal stimulation. However, cytoarchitecture, axonal transmission latency and synaptic delays may all play roles in determining this timing, and other monosynaptic connections are not readily reflected through polysynaptic oPSCs latency measurements. This highlights the importance of using opsin-assisted monosynaptic mapping rather than timing to identify the details of synaptic connectivity.

While our opsin-assisted circuit mapping provides us with a new level of insight into HVC synaptic circuitry, there are limitations to this research that should be considered. All circuit mapping in this study was carried out in brain slices from adult male zebra finches. Future studies will be needed to examine how this adult connectivity pattern relates to patterns of connectivity in juveniles during sensory or sensorimotor phases of vocal learning and connectivity patterns in female birds. Ex-vivo brain slices have historically been demonstrated to be a reliable window into neuronal and circuit function. However, intrinsic activity patterns of some neuronal subtypes are known to change in acute slices (*Opitz et al., 2017*). Moreover, the slicing procedure severs both neuromodulatory and neurotransmitter afferents, potentially attenuating or removing synaptic inputs to some circuits (*Ballanyi and Ruangkittisakul, 2009*). Our axonal terminal optical manipulation recruits all the fibers expressing eGtACR1 under the cone of light of the microscope objective. Therefore, our estimates of postsynaptic responses and likelihood of finding responses are based on the simultaneous release of neurotransmitter from all the afferent terminals expressing eGtACR1, and they should be understood in this context. Future studies will need to employ conditions of minimal stimulation that may further characterize the specific properties of monosynaptic and polysynaptic neurotransmission in the examined inputs. Lastly, the amplitude and likelihood of recording oPSCs directly depend on the expression rate and tropism of our viruses, and this may differ across the four afferent pathways studied here. We report low levels of viral expression in Av, which may skew our results. However, by testing oPSCs in all three $HVC_{PN}$ classes in each bird help ensure that the relative contribution of Av afferent inputs onto different classes of HVC neurons is not simply an artifact associated with the efficiency of viral transduction.

Although a complete description of HVC circuitry will require the examination of other potential inputs (i.e. $RA_{HVC}$ PNs, NCM, A11 glutamatergic neurons *Roberts et al., 2008*; *Ben-Tov et al., 2023*; *Louder et al., 2024*) and a characterization of interneuron synaptic connectivity; here, we provide a map of the synaptic connections between the four best described afferents to HVC and its three populations of projection neurons in adult birds. These results provide a view into the synaptic

underpinnings of circuits fundamental for the learning and production of vocalizations. Moreover, they reveal essential connectivity patterns that can underlie song motor control and learning of syllable order. This picture of input-output organization spanning sensory, thalamic, and premotor areas may thus offer insights into the circuits for human language learning and production. Lastly, the opsin-assisted circuit mapping strategy we used provides an approach to fully characterize other long-range and local circuitry in the songbird brain, which will be critical for understanding and modeling the remarkable vocal imitation abilities of songbirds.

## Materials and methods

### Animals

Experiments described in this study were conducted using adult male zebra finches (130–500 days post hatch). All procedures were performed in accordance with protocol 2016–101562 G approved by the Animal Care and Use Committee at UT Southwestern Medical Center.

### Viral vectors

The following adeno-associated viral vectors were used in these experiments: rAAV2/9/CBh-FLP and rAAV-CBh-eGtACR1-mScarlet (IDDRC Neuroconnectivity Core at Baylor College of Medicine or UT Southwestern AAV Viral Vector Core). All viral vectors were aliquoted and stored at –80 °C until use.

### Constructs

CBh-eGtACR1-mScarlet was generated by inserting KpnI and AgeI restriction sites flanking EF1a in the pAAV-EF1a-F-FLEX-mNaChBac-T2A-tdTomato (a gift from Dr. Massimo Scanziani; Addgene plasmid # 60658) through PCR cloning. The CBh promoter was isolated from pX330-U6-Chimeric_BB-CBh-hSpCas9 (a gift from Dr. Feng Zhang; Addgene plasmid # 42230) by digestion of the KpnI and AgeI restriction sites and inserted into the KpnI/AgeI restriction sites of the pAAV-EF1a-F-FLEX-mNaChBac-T2A-tdTomato backbone. Next, we isolated mScarlet from pmScarlet_C1 (a gift from Dr. Dorus Gadella; Addgene plasmid # 85042) through PCR amplification, inserting a BamHI restriction site into the 5′ end of the ORF. We digested the amplified fragments with BamHI and BsrGI, inserting the digested fragments into BamHI/BsrGI restriction sites of the pAAV-CBh-F-FLEX-mNaChBac-T2A-tdTomato backbone. eGtACR1 was then amplified from pAAV-CAG-DIO-NLS-mRuby3-IRES-eGtACR1-ST (a gift from Dr. Hillel Adesnik; Addgene plasmid # 109048) with a BmtI restriction site on the 5′ end and an AflII restriction site on the 3′ end, avoiding amplification of the soma-targeting sequence in the original plasmid. We then digested the amplified fragment with BmtI and AflII, inserting the fragment into the BmtI/AflII sites of the pAAV-CBh-F-FLEX-mNaChBac-T2A-mScarlet.

For CBh- Flpo, we amplified the FLP ORF from pCAG- Flpo (a gift from Dr. Massimo Scanziani; Addgene plasmid # 60662) with an AgeI restriction site on the 5′ end. We then digested the amplified fragment with AgeI and EcoRI, inserting the fragment into the AgeI/EcoRI sites of the CBh-eGtACR1-mScarlet.

Constructs were verified with Sanger sequencing.

### Stereotaxic surgery

All surgical procedures were performed under aseptic conditions. Birds were anaesthetized using isoflurane inhalation (0.8–1.5%) and placed in a stereotaxic surgical apparatus. The centers of HVC, Nlf, mMAN, and RA were identified with electrophysiological recordings, and Area X, Av, and Uva were identified using stereotaxic coordinates (approximate stereotaxic coordinates relative to inter-aural zero and the brain surface: head angle, rostral-caudal, medial-lateral, dorsal-ventral (in mm). HVC: 45°, AP 0, ML ±2.4, DV –0.2–0.6; Nlf: 45°, AP 1.75, ML ±1.75, DV –2.4–1.8; mMAN: 20°, AP 5.1, ML ±0.6, DV –2.1–1.6; RA: 70°, AP –1.5, ML ±2.5, DV –2.4–1.8; X: 45°, AP 4.6, ML ±1.6, DV –3.3–2.7; Av: 45°, AP 1.65, ML ±2.0, DV –0.9; UVA: 20°, AP 2.8, ML ±1.6, DV –4.8–4.2).

Viral injections were performed using previously described procedures (*Roberts et al., 2017*; *Roberts et al., 2012*). Briefly, a cocktail of adeno-associated viral vectors (1:2 of rAAV-CBh-FLP and rAAV-DIO-CBh-eGtACR1, respectively) was injected into Nlf, Uva, mMAN, or Av (1 nl/s, for a total of 1.5 µl/hemisphere). A minimum of 4 weeks after viral injections, fluorophore-conjugated retrograde tracers (Dextran 10,000 MW, AlexaFluor 488 and 568, Invitrogen; FastBlue, Polysciences) were

injected bilaterally into Area X, Av, and RA. Tracer injections (160 nl, 5x32 nl, 32 nl/s every 30 s) were performed using previously described procedures (*Roberts et al., 2017*; *Roberts et al., 2012*; *Xiao et al., 2018*).

## In vivo extracellular recordings

To test the functional expression of eGtACR1 in HVC afferent pathways, we performed extracellular recording of HVC activity in anesthetized birds previously injected with rAAV-DIO-CBh-eGtACR1 and rAAV-CBh-FLP (2:1). We performed the recordings under light isoflurane anesthesia (0.8%) with Carbostar carbon electrodes (impedance: 1670 microhms/cm; Kation Scientific). During the recordings, we lowered a 400 µm multimodal optical fiber to the brain surface overlaying HVC and delivered light stimulation (470 nm, ≈ 20 mW, 1 s). Signals were acquired at 10 KHz and filtered (high-pass 300 Hz, low-pass 20 KHz). We used Spike2 to analyze the spike rate (binned every 10ms) and build a peri-stimulus time histogram to evaluate the effect of light stimulation across trials (5–15 trials/hemisphere). We sampled a minimum of one site and a maximum of three HVC sites/hemisphere. Birds with weak or no optically evoked responses were excluded from further experiments.

## Ex vivo physiology

### Slice preparation

Zebra finches were deeply anesthetized with isoflurane and decapitated. The brain was removed from the skull and submerged in cold (1–4°C) oxygenated dissection buffer. Acute sagittal 230 µm brain slices were cut in ice-cold carbogenated (95% O2/5% CO2) solution, containing (in mM) 110 choline chloride, 25 glucose, 25 NaHCO3, 7 MgCl2, 11.6 ascorbic acid, 3.1 sodium pyruvate, 2.5 KCl, 1.25 NaH2PO4, 0.5 CaCl2; 320–330 mOsm. Individual slices were incubated in a custom-made holding chamber filled with artificial cerebrospinal fluid (aCSF), containing (in mM): 126 NaCl, 3 KCl, 1.25 $NaH_2PO_4$, 26 $NaHCO_3$, 10 D-(+)-glucose, 2 $MgSO_4$, 2 $CaCl_2$, 310 mOsm, pH 7.3–7.4, aerated with a 95% $O_2$/5% $CO_2$ mix. Slices were incubated at 36 °C for 20 min, and then kept at RT for a minimum of 45 min before recordings.

### Slice electrophysiological recording

Slices were constantly perfused in a submersion chamber with 32 °C oxygenated normal aCSF. Patch pipettes were pulled to a final resistance of 3–5 MΩ from filamented borosilicate glass on a Sutter P-1000 horizontal puller. HVC-PN classes, as identified by retrograde tracers, were visualized by epifluorescence imaging using a water immersion objective (×40, 0.8 numerical aperture) on an upright Olympus BX51 WI microscope, with video-assisted infrared CCD camera (Q-Imaging Rolera). Data were low-pass filtered at 10 kHz and acquired at 10 kHz with an Axon MultiClamp 700B amplifier and an Axon Digidata 1550B Data Acquisition system under the control of Clampex 10.6 (Molecular Devices).

For voltage clamp whole-cell recordings, the internal solution contained (in mM): 120 cesium methanesulfonate, 10 CsCl, 10 HEPES, 10 EGTA, 5 Creatine Phosphate, 4 ATP-Mg, 0.4 GTP-Na (adjusted to pH 7.3–7.4 with CsOH).

Optically evoked post-synaptic currents (oPSCs) were measured by delivering 2 light pulses (1ms, spaced 50ms, generated by a CoolLED *p*E300) focused on the sample through the 40 X immersion objective. Sweeps were delivered every 10 s. Synaptic responses were monitored while holding the membrane voltage at –70 mV (for excitatory oPSCs (oEPSCs)) and +10 mV (for inhibitory oPSCs (oIPSCs)). The light stimulation mostly elicited excitatory currents with amplitude in the range of ~50–600 pA. We calibrated the light stimulation intensity to ~50% of the maximum amplitude. If the amplitude exceeded 600 pA, we further reduced the light intensity. In cases with currents below 50 pA, we set 20 pA as the minimum cut-off current amplitude in order to proceed with experiments. When both oIPSC and oEPSC were measured, the measurement was conducted at the same light stimulation intensity to allow direct comparison. Access resistance (10–30 MΩ) was monitored throughout the experiment, and for reliability of amplitude measurement, recordings where it changed more than 20% were discarded from further analysis. We calculated the paired-pulse ratio (PPR) as the amplitude of the second peak divided by the amplitude of the first peak elicited by the twin stimuli. However, due to the slow kinetics of eGtACR1, the results would be difficult to interpret, and therefore we are not currently reporting them. The excitatory-inhibitory (E/I) ratio was calculated by dividing the

amplitude of the oEPSC at –70 mV by the amplitude of the oIPSC at +10 mV, while stimulating at identical light intensities. To validate inhibitory and excitatory oPSCs as GABAergic and gluta-matergic, respectively, we bath applied the GABAa receptor antagonist SR 95531 hydrobromide (gabazine, µM) while holding the cell at +10 mV, or the AMPA receptor antagonist 6,7-Dinitroquin oxaline-2,3-dione (DNQX, µM) while holding the cell at –70 mV. In another subset of cells, once the baseline measures were established, we tested for monosynaptic connectivity. To isolate monosyn-aptic currents driven by optogenetic stimulation, we bath-applied 1 µM Tetrodotoxin (TTX), followed by 100 µM 4-Aminopyridine (4AP) and measured the amplitude of oPSCs returning following 4-AP application. Currents under 5 pA were considered not reliable, based on the signal to noise of the recordings, and were assigned value 0 for further analysis. Therefore, average amplitudes and plots relative to oPSCs amplitudes (panels C, D of *Figures 2, 3, 5 and 6*) report only cells whose values surpass this 5 pA threshold. oPSCs = 0 pA were considered to calculate the rate of success in evoking oPSCs and to evaluate whether currents are monosynaptic following TTX and 4AP application. For the sake of classification, any current rescued by 4AP application with an amplitude lower than 10 pA was considered non-monosynaptic. Birds where no oPSC was recorded in any $HVC_{PN}$ throughout the experimental day were excluded from further analysis.

## Histology

After electrophysiological recordings, the slices were incubated in 4% PFA in PBS. Sections were then washed in PBS, mounted on glass slides with Fluoromount-G (eBioscience, CA, USA), and visualized under an LSM 880 laser-scanning confocal microscope (Carl Zeiss, Germany).

## In situ hybridization

We used the hairpin chain reaction system from Molecular Instruments as previously described (*Ben-Tov et al., 2023*). Briefly, 5 days after having injected HVC with the retrograde tracer Fast Blue (150 nl/HVC, Polysciences), the brain was perfused with 4% paraformaldehyde (PFA), post-fixed at 4 °C for 12 hr, and dehydrated in 30% sucrose/4% PFA at 4 °C for 24 hr. The dehydrated brains were sagittally cryosectioned at 40 µm and collected into ice-cold 4% PFA. Sections were then incubated twice in PBS for 3 min, in 5% SDS/PBS for 45 min, three times in 2 X SSCT for 15 min, in Hybridization Buffer for 5 min, and finally in 10 nM each probe/Hybridization Buffer at 37 °C for 24 hr. The sections were then washed four times in Probe Wash Buffer for 15 min at 37 °C, three times in 2 X SSCT for 15 min at room temperature, then incubated in Amplification Buffer for 30 min at room temperature. Each Alexa fluor-conjugated hairpin was denatured in separated PCR tubes at 95 °C for 90 s and cooled down to room temperature at 2 °C/s. Sections were incubated in solutions containing 36 nM of each hairpin in Amplification Buffer at room temperature for 24 hr. The sections were then washed four times with 2 X SSCT for 15 min and mounted with mounting medium.

## Experimental design and analysis

Electrophysiological data were analyzed with Clampfit (Molecular Devices). All data were tested for normality using the Shapiro-Wilk Test. Parametric and non-parametric statistical tests were used as appropriate. One-way ANOVA or the Kruskal-Wallis test was performed when comparisons were made across more than two conditions. Two-Way ANOVA or Mixed-effect analysis was used to compare multiple conditions across different groups. Fisher's exact tests were used to compare the probability of finding optically evoked responses. Statistical significance refers to *$p<0.05$, ** $p<0.01$, *** $p<0.001$.

The Sankey diagram in *Figure 7C* was drawn with SankeyMATIC (https://www.sankeymatic.com/) setting ribbons proportional to the percentage of interrogated synaptic connections between each afferent area and HVCPN class displaying excitatory currents or lack thereof. From the hubs reporting existing synaptic connections (center), ribbons are scaled based on the percentage of monosynaptic and polysynaptic connectivity for each combination.

## Acknowledgements

We thank Drs. David Perkel, Frank Meye, Salvatore Lecca and Manuel Mameli for discussion and comments on an initial version of the manuscript. We thank Andrea Guerrero, Luis Garcia, and Jennifer

Holdway for laboratory support. This research was supported by grants from the US National Institutes of Health UF1NS115821 and R01NS108424 to TFR. DA was supported by F99NS124172.

## Additional information

### Funding

| Funder | Grant reference number | Author |
|---|---|---|
| National Institute of Neurological Disorders and Stroke | UF1NS115821 | Todd F Roberts |
| National Institute of Neurological Disorders and Stroke | R01NS108424 | Todd F Roberts |
| National Institute of Neurological Disorders and Stroke | F99NS124172 | Danyal H Alam |

The funders had no role in study design, data collection and interpretation, or the decision to submit the work for publication.

### Author contributions

Massimo Trusel, Conceptualization, Formal analysis, Validation, Investigation, Visualization, Methodology, Writing – original draft, Writing – review and editing; Ziran Zhao, Formal analysis, Investigation, Methodology; Danyal H Alam, Resources, Investigation, Methodology; Ethan S Marks, Formal analysis, Investigation; Maaya Z Ikeda, Investigation, Visualization; Todd F Roberts, Conceptualization, Supervision, Funding acquisition, Project administration, Writing – review and editing

### Author ORCIDs

Massimo Trusel ⓘ https://orcid.org/0000-0001-6208-2476
Todd F Roberts ⓘ https://orcid.org/0000-0002-0967-6598

### Ethics

Experiments described in this study were conducted using adult male zebra finches (130-500 days post hatch). All procedures were performed in accordance with protocol 2016-101562-G approved by Animal Care and Use Committee at UT Southwestern Medical Center.

Reviewer #1 (Public review): https://doi.org/10.7554/eLife.104609.3.sa1
Reviewer #2 (Public review): https://doi.org/10.7554/eLife.104609.3.sa2
Reviewer #3 (Public review): https://doi.org/10.7554/eLife.104609.3.sa3
Author response https://doi.org/10.7554/eLife.104609.3.sa4

## Additional files

### Supplementary files

MDAR checklist

### Data availability

All data generated or analysed during this study are included in the manuscript and supporting files. The source data used in this study have been uploaded to UT Southwestern Research Data Repository (https://doi.org/10.18738/T8/FFAM6F).

The following dataset was generated:

| Author(s) | Year | Dataset title | Dataset URL | Database and Identifier |
|---|---|---|---|---|
| Trusel M | 2025 | Electrophysiology Data from Trusel et al., eLife 2025 | https://doi.org/10.18738/T8/FFAM6F | UT Southwestern Research Data Repository, 10.18738/T8/FFAM6F |

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
