## [Editor Report · eLife Assessment]

The songbird vocal motor nucleus HVC contains cells that project to the basal ganglia, the auditory system, or to downstream vocal motor structures. In this **fundamental** study, the authors conduct optogenetic circuit mapping to clarify how four distinct inputs to HVC act on these distinct HVC cell types. They provide **compelling** evidence that all long range projections engage inhibitory circuits in HVC and can also exhibit cell-type specific preferences in monosynaptic input strength. Understanding HVC at this microcircuit level is critical for constraining models of song learning and production.

---

## [Referee Report · Reviewer #1 (Public review)]

Summary:

This work has crated the map of synaptic connectivity between the inputs and outputs of song premotor nucleus, HVC in zebra finches to understand how sensory (auditory) to motor circuit interact to coordinate song production and learning. The authors optimized the optogenetic technique via AAV to manipulate auditory inputs from a specific auditory area one-by-one and recorded synaptic activity from a neuron in HVC with whole-cell recording from slice preparation with identification of projection area by retrograde neuronal tracing. These thorough and detailed analysis provide compelling evidence of synaptic connections between 4 major auditory inputs (3 forebrain and 1 thalamic regions) within three projection neurons in the HVC; all areas give monosynaptic excitatory inputs and polysynaptic inhibitory inputs, but proportions of projection to each projection neuron varied. They also find specific reciprocal connections between mMAN and Av. Taken together the authors provide the map of synaptic connection between intercortical sensory to motor areas which is suggested to be involved in zebra finch song production and learning.

Strengths:

The authors optimized optogenetical tools with eGtACR1 by using AAV which allow them to manipulate synaptic inputs in a projection-specific manner in zebra finches. They also identify HVC cell type based on projection area. With their technical advance and thorough experiments, they provided detailed map synaptic connection and gave insights into the neuronal circuit for auditory guided vocal (motor) learning.

Weaknesses:

As this study is in adult brain slices, there might be a gap to the functions in developmental song learning.

---

## [Referee Report · Reviewer #2 (Public review)]

Summary:

The manuscript describes synaptic connectivity in Songbird cortex four main classes of sensory neurons afferents onto three known classes of projection neurons of the pre-motor cortical region HVC. HVC is a region associated with the generation of learned bird song. Investigators here use all male zebra finches to examine the functional anatomy of this region using patch clamp methods combined with optogenetic activation of select neuronal groups.

Strengths:

The quality of the recordings is extremely high and the quantity of data is on a very significant scale, this will certainly aid the field.

Weaknesses:

Could make the figures a little easier to navigate by having some atlas drawings.

Comments on revisions:

The authors have addressed the minor concerns and suggestions

---

## [Referee Report · Reviewer #3 (Public review)]

Nucleus HVC is critical both for song production as well as learning and arguably, sitting at the top of the song control system, is the most critical node in this circuit receiving a multitude of inputs and sending precisely timed commands that determine the temporal structure of song. The complexity of this structure and its underlying organization seem to become more apparent with each experimental manipulation, and yet our understanding of the underlying circuit organization remains relatively poorly understood. In this study, Trusel and Roberts use classic whole-cell patch clamp techniques in brain slices coupled with optogenetic stimulation of select inputs to provide a careful characterization and quantification of synaptic inputs into HVC. By identifying individual projections neurons using retrograde tracer injections combined with pharmacological manipulations, they classify monosynaptic inputs onto each of the three main classes of glutamatergic projection neurons in HVC (RA-, Area X- and Av-projecting neurons). This study is remarkable in the amount of information that it generates, and the tremendous labor involved for each experiment, from the expression of opsins in each of the target inputs (Uva, NIf, mMAN and Av), the retrograde labelling of each type of projection neuron, and ultimately the optical stimulation of infected axons while recording from identified projection neurons. Taken together, this study makes an important contribution to increasing our identification, and ultimately understanding, of the basic synaptic elements that make up the circuit organization of HVC, and how external inputs, which we know to be critical for song production and learning, contribute to the intrinsic computations within this critic circuit.

This study is impressive in its scope, rigorous in its implementation and thoughtful regarding its limitations. The manuscript is well written, and I appreciate the clarity with which the authors use our latest understanding of the evolutionary origins of this circuit to place these studies within a larger context and their relevance to the study of vocal control, including human speech. My comments are minor and primarily about legibility, clarification of certain manipulations and organization of some of the summary figures.

Comments on revisions:

The authors have done a very nice job addressing the reviewers' comments.

---

## [Author Response]

The following is the authors’ response to the original reviews

**Public Reviews:**

**Reviewer #1 (Public review):**
Summary:This work tried to map the synaptic connectivity between the inputs and outputs of the song premotor nucleus, HVC in zebra finches to understand how sensory (auditory) to motor circuits interact to coordinate song production and learning. The authors optimized the optogenetic technique via AAV to manipulate auditory inputs from a specific auditory area one-by-one and recorded synaptic activity from a neuron with whole-cell recording from slice preparation with identification of the projection area by retrograde neuronal tracing. This thorough and detailed analysis provides compelling evidence of synaptic connections between 4 major auditory inputs (3 forebrain and 1 thalamic region) within three projection neurons in the HVC; all areas give monosynaptic excitatory inputs and polysynaptic inhibitory inputs, but proportions of projection to each projection neuron varied. They also find specific reciprocal connections between mMAN and Av. Taken together the authors provide the map of the synaptic connection between intercortical sensory to motor areas which is suggested to be involved in zebra finch song production and learning.Strengths:The authors optimized optogenetic tools with eGtACR1 by using AAV which allow them to manipulate synaptic inputs in a projection-specific manner in zebra finches. They also identify HVC cell types based on projection area. With their technical advance and thorough experiments, they provided detailed map synaptic connections.Weaknesses:As it is the study in brain slice, the functional implication of synaptic connectivity is limited. Especially as all the experiments were done in the adult preparation, there could be a gap in discussing the functions of developmental song learning.

We thank the reviewer for their appreciation of our work. Although we agree that there can be limitations to brain slice preparations, the approaches used here for synaptic connectivity mapping are well-designed to identify long-range synaptic connectivity patterns. Optogenetic stimulation of axon terminals in brain slices does not require intact axons and works well when axons are cut, allowing identification of all inputs expressing optogenetic channels from aXerent regions. Terminal stimulation in slices yields stable post-synaptic responses for hours without rundown, assuring that polysynaptic and monosynaptic connections can be reliably identified in our brain slices. Additionally, conducting similar types of experiments in vivo can run into important limitations. First, the extent of TTX and 4-AP diXusion, which is necessary for identification of long-range monosynaptic connections, can be diXicult to verify in vivo - potentially confounding identification of monosynaptic connectivity. Second, conducting whole-cell patch-clamp experiments in vivo, particularly in deeper brain regions, is technically challenging, and would limit the number of cells that can be patched and increase the number of animals needed.

We agree that there may well be important diXerences between adult connectivity and connectivity patterns in the juvenile brain. Indeed, learning and experience during development almost certainly shape connectivity patterns and these patterns of connectivity may change incrementally and/or dynamically during development. Ultimately, adult connectivity patterns are the result of changes in the brain that accrue over development. Given that this is the first study mapping long-range connectivity of HVC input-output pathways, we reasoned that the adult connectivity would provide a critical reference allowing future studies to map diXerent stages of juvenile connectivity and the changes in connectivity driven by milestones like forming a tutor song memory, sensorimotor learning, and song crystallization.

In this revision we worked to better highlight the points raised above and thank the reviewer for their comments.

**Reviewer #2 (Public review):**
Summary:The manuscript describes synaptic connectivity in the Songbird cortex's four main classes of sensory neuron aXerents onto three known classes of projection neurons of the pre-motor cortical region HVC. HVC is a region associated with the generation of learned bird songs. Investigators here use all male zebra finches to examine the functional anatomy of this region using patch clamp methods combined with optogenetic activation of select neuronal groups.Strengths:The quality of the recordings is extremely high and the quantity of data is on a very significant scale, this will certainly aid the field.Weaknesses:The authors could make the figures a little easier to navigate. Most of the figures use actual anatomical images but it would be nice to have this linked with a zebra finch atlas in more of a cartoon format that accompanied each fluro image. Additionally, for the most part, figures showing the labeling lack scale bar values (in um). These should be added not just shown in the legends.The authors could make it clear in the abstract that this is all male zebra finches - perhaps this is obvious given the bird song focus, but it should be stated. The number of recordings from each neuron class and the overall number of birds employed should be clearly stated in the methods (this is in the figures, but it should say n=birds or cells as appropriate).The authors should consider sharing the actual electrophysiology records as data.

We thank the reviewer for their assessment of our research and suggestions. We have implemented many of these suggestions and provide details in our response to their specific Recommendations. Additionally, we are organizing our data and will make it publicly available with the version of record.

**Reviewer #3 (Public review):**
Nucleus HVC is critical both for song production as well as learning and arguably, sitting at the top of the song control system, is the most critical node in this circuit receiving a multitude of inputs and sending precisely timed commands that determine the temporal structure of song. The complexity of this structure and its underlying organization seem to become more apparent with each experimental manipulation, and yet our understanding of the underlying circuit organization remains relatively poorly understood. In this study, Trusel and Roberts use classic whole-cell patch clamp techniques in brain slices coupled with optogenetic stimulation of select inputs to provide a careful characterization and quantification of synaptic inputs into HVC. By identifying individual projection neurons using retrograde tracer injections combined with pharmacological manipulations, they classify monosynaptic inputs onto each of the three main classes of glutamatergic projection neurons in HVC (RA-, Area X- and Av-projecting neurons). This study is remarkable in the amount of information that it generates, and the tremendous labor involved for each experiment, from the expression of opsins in each of the target inputs (Uva, NIf, mMAN, and Av), the retrograde labelling of each type of projection neuron, and ultimately the optical stimulation of infected axons while recording from identified projection neurons. Taken together, this study makes an important contribution to increasing our identification, and ultimately understanding, of the basic synaptic elements that make up the circuit organization of HVC, and how external inputs, which we know to be critical for song production and learning, contribute to the intrinsic computations within this critic circuit.This study is impressive in its scope, rigorous in its implementation, and thoughtful regarding its limitations. The manuscript is well-written, and I appreciate the clarity with which the authors use our latest understanding of the evolutionary origins of this circuit to place these studies within a larger context and their relevance to the study of vocal control, including human speech. My comments are minor and primarily about legibility, clarification of certain manipulations, and organization of some of the summary figures.

We thank the reviewer for their thoughtful assessment of our research.

**Recommendations for the authors:**
The following recommendations were considered by all reviewers to be important to incorporate for improving this paper:(1) Clarify the site of viral injection and the possibility of labeling other structures (a) Show images of viral injection sites.

We provide a representative image of viral expression for each pathway studied in this manuscript. Please see panel A in Figures 2-3 and 5-6 showing our viral expression in Uva, NIf, mMAN, and Av respectively.

b) Include in discussion caveats that the virus may spread beyond the boundaries of structures (e.g. especially injections into NIF could spread into Field L).

For each HVC aXerent nucleus we have now included a sentence describing the possible spread of viral infection in surrounding structures in the Results. We also now expanded the image from the Av section to include NIf, to showcase lack of viral expression in NIf (see Fig. 6A).

(2) Clarify the logic and precise methods of the TTX and 4-AP experimentsa) Please see the detailed issue raised by Reviewer 3, Major Point 1 below.

The TTX and 4AP application is the gold-standard of opsin-assisted synaptic circuit interrogation, pioneered by the Svoboda lab in 2009 (Petreanu, Mao et al. 2009) and widely used to assess monosynaptic connectivity in multiple brain circuits, as summarized in a recent review(Linders, Supiot et al. 2022). We now better describe the logic of this approach in the second paragraph of the Results section and cite the first description of this method from the Svoboda lab and a recent review weighing this method with other optogenetic methods for tracing synaptic connections in the brain.

(3) Include caveats in discussiona) Note that there may be other inputs to HVC that were not examined in this study (e.g. CMM, Field L)

In our original manuscript we did state “Although a complete description of HVC circuitry will require the examination of other potential inputs (i.e. RA_HVC_ PNs, A11 glutamatergic neurons(Roberts, Klein et al. 2008, Ben-Tov, Duarte et al. 2023)) and a characterization of interneuron synaptic connectivity, here we provide a map of the synaptic connections between the 4 best described aPerents to HVC and its 3 populations of projection neurons” in the last paragraph of the Discussion. We have now edited this sentence to include the projection from NCM to HVC and cited Louder et al., 2024.

We have extensively mapped input pathways to HVC, and consistent with Vates (Vates, Broome et al. 1996) we have not found evidence that Field L projects to HVC. Rather that it projects to the shelf region outside of HVC. Consistent with this, we do not see retrogradely labeled neurons in Field L following tracer injections confined to HVC (see Fig. 3G). Additionally, we find that CM projections to HVC arise from the nucleus Avalanche (Roberts, Hisey et al. 2017) which we specifically examine in this study. We do not dispute that there may be other pathways projecting to HVC that will need to be examined in the future, including known projections from neuromodulatory regions and RA, from developmentally restricted pathway(s) like NCM (Louder, Kuroda et al. 2024), and from yet unidentified pathways.

b) Also note that birds in this study were adults and that some inputs to HVC likely to be important for learning may recede during development (e.g. Louder et al, 2024).

In the second to last paragraph of the Discussion we now state: While our opsin-assisted circuit mapping provides us with a new level of insight into HVC synaptic circuitry, there are limitations to this research that should be considered. All circuit mapping in this study was carried out in brain slices from adult male zebra finches. Future studies will be needed to examine how this adult connectivity pattern relates to patterns of connectivity in juveniles during sensory or sensorimotor phases of vocal learning and connectivity patterns in female birds.

(4) Consider cosmetic changes to figures as suggested by Reviewers 2-3 below.

We thank the reviewers for their suggestions and have implemented the changes as best we can.

(5) Address all minor issues raised below.
**Reviewer #1 (Recommendations for the authors):**
I see this study is well designed to answer the author's specific question, mapping synaptic auditorymotor connections within HVC. Their experiments with advanced techniques of projection-specific optogenetic manipulation of synaptic inputs and retrograde identification of projection areas revealed input-output combination selective synaptic mapping.As I found this study advanced our knowledge with the compelling dataset, I have only some minor comments here.(1) One technical concern is we don't see how much the virus infection was focused on the target area and if we can ignore the eXect of synaptic connectivity from surrounding areas. As the amount of virus they injected is large (1.5ul) and target areas are small, we assume the virus might spread to the surrounding area, such as field L which also projects to HVC when targeting Nif. While I think the majority of the projections were from their target areas, it would be better to mention (also the images with larger view areas) the possibilities of projections of surrounding areas.

We agree with the reviewer about the concern about specificity of viral expression. For this reason, we included sample images of the viral expression in each target area (panel A in Fig. 2,3,5,6). We have now also included a sentence at the beginning of each subsection of our Result to describe how we have ensured interpretability of the results. Uva and mMAN’s surrounding areas are not known to project to HVC. Possible cross-infection is an issue for Av and NIf, and we checked each bird’s injection site to ensure that eGtACR1+ cells were not visible in the unintended HVC-projecting areas.

As mentioned in our response the public comment, consistent with Vates (Vates, Broome et al. 1996) we do not see evidence that Field L projects directly to HVC (see Fig. 3G).

(2) Another concern about the technical issue is the damage to axonal projections. While I understand the authors stimulated axonal terminals axonal projections were assumed to be cut and their ability to release neurotransmitters would be reduced especially after long-term survival or repeated stimulation. Mentioning whether projection pathways were within their 230um-thick slice (probably depends on input sites) or not and the eXect of axonal cut would be helpful.

We agree that slice electrophysiology has limitations. However, we disagree with the claim of reduced reliability or stability of the evoked response. We and others find that electrical and optogenetic repeated terminal stimulation in slices can yield stable post-synaptic responses for tens of minutes and even hours (Bliss and Gardner-Medwin 1973, Bliss and Lomo 1973, Liu, Kurotani et al. 2004, Pastalkova, Serrano et al. 2006, Xu, Yu et al. 2009, Trusel, Cavaccini et al. 2015, Trusel, Nuno-Perez et al. 2019). Indeed, long-term synaptic plasticity experiments in most preparations and across brain areas rely on such stability of the presynaptic machinery for synaptic release, despite axons being severed from their parent soma. Our assumption is the vast majority, if not all, connections between axon terminals and their cell body in the aXerent regions have been cut in our preparations. Nonetheless, the diversity of outcomes we report (currents returning after TTX+4AP or not, depending on the specific combination of input and HVCPN class) is consistent with the robustness of the synaptic interrogation method.

(3) While I understand this study focused on 4 major input areas and the authors provide good pictures of synaptic HVC connections from those areas, HVC has been reported to receive auditory inputs from other areas as well (CMM, FieldL, etc.). It is worth mentioning that there are other auditory inputs and would be interesting to discuss coordination with the inputs from other areas.

We have extensively mapped input pathways to HVC, and consistent with Vates (Vates, Broome et al. 1996) we have not found evidence that Field L projects to HVC. Rather that it projects to the shelf region outside of HVC. Consistent with this, we do not see retrogradely labeled neurons in Field L following tracer injections confined to HVC (see Fig. 3G). Additionally, we find that CM projections to HVC arise from the nucleus Avalanche (Roberts, Hisey et al. 2017) which we specifically examine in this study. We do not dispute that there may be other pathways projecting to HVC that will need to be examined in the future, including known projections from neuromodulatory regions and RA, from developmentally restricted pathway(s) like NCM (Louder, Kuroda et al. 2024), and from yet unidentified pathways.

(4) The HVC local neuronal connections have been reported to be modified and a recent study revealed the transient auditory inputs into HVC during song learning period. The author discusses the functions of HVC synaptic connections on song learning (also title says synaptic connection for song learning), however, the experiments were done in adults and dp not discuss the possibility of diXerent synaptic connection mapping in juveniles in the song learning period. Mentioning the neuronal activities and connectivity changes during song learning is important. Also, it would be helpful for the readers to discuss the potential diXerences between juveniles/adults if they want to discuss the functions of song learning.

We now mention in the Discussion that this is an important caveat of our research and that future studies will be needed to examine how these adult connectivity patterns relate to connectivity patterns in juveniles during sensory or sensorimotor phases of vocal learning and connectivity patterns in female birds. Nonetheless, the title and abstract cite song learning because it is important for the broader public to understand that at least some of these aXerent brain regions carry an essential role in song learning (Foster and Bottjer 2001, Roberts, Gobes et al. 2012, Roberts, Hisey et al. 2017, Zhao, Garcia-Oscos et al. 2019, Koparkar, Warren et al. 2024).

**Reviewer #2 (Recommendations for the authors):**
The work is very detailed and will be an important resource to those working in the field. The recordings are of a high quality and lots of information is included such as measures of response kinetics amplitude and pharmacological confirmation of excitatory and inhibitory synaptic responses. In general, I feel the quality is extremely high and the quantity of data is on a very significant exhaustive scale that will certainly aid the field. I have come at this conclusion as a non zebra finch person but I feel the connection information shown will be of benefit given its high quality.Figure 7 is a nice way of showing the overall organization. Optional suggestion, consider highlighting anything in Figure 7 that results in a new understanding of the song system as compared to previous work on anatomy and function.

We thank the reviewer for the kind comments about our research. We have highlighted our newly found connection between mMAN and Av and all the connections onto the HVC PNs in Panel B are newly identified in this study.

**Reviewer #3 (Recommendations for the authors):**
Major points(1) Clarification regarding methods for determining monosynaptic events:One of the manipulations that I struggled the most with was those describing the use of TTX + 4AP to isolate monosynaptic events. Initially, not being as familiar with the use of optically based photostimulation of axons to release transmitter locally, I was initially confused by statements such as "we found that oEPSC returned after application of TTX+4AP". This might be clear to someone performing these manipulations, but a bit more clarification would be helpful. Should I assume that an existing monosynaptic EPSC would be masked by co-occurring polysynaptic IPSCs which disappear following application of TTX + 4AP, thereby unmasking the monosynaptic EPSC, thereby causing the EPSC to "return"? A word that I am not sure works. Continuing my confusion with these experiments, I am unsure how this cocktail of drugs is added, if it is even added as a cocktail, which is what I initially assumed. The methods and the results are not so clear if they are added in sequence and why and if traces are recorded after the addition of both drugs or if they are recorded for TTX and then again for TTX + 4AP. Finally, looking at the traces in the experimental figures (e.g. Figures 2F, 3F, 5F, and 6F), it is diXicult to see what is being shown, at least for me. First, the authors need to describe better in the results why they stimulate twice in short succession and why they seem to use the response to the second pulse (unless I am mistaken) to measure the monosynaptic event. Second, I was confused by the traces (which are very small) in the presence of TTX. I would have expected to see a response if there was a monosynaptic EPSC but I only seem to see a flat line.The confusion that I list above might be due in part to my ignorance, but it is important in these types of papers not to assume too much expertise if you want readers with a less sophisticated understanding of synaptic physiology to understand the data. In other words, a little bit more clarity and hand-holding would be welcome.

We understand the reviewer’s confusion about the methodology. In Voltage clamp, the amplifier injects current through the electrode maintaining the membrane voltage to -70mV, where the equilibrium potential for Cl- is near equilibrium, and therefore the only synaptic current evoked by light stimulation is due to cation influx, mainly through AMPA receptors (see Fig. 1). Therefore, cooccurring polysynaptic IPSCs wouldn’t be visible. We examine those holding the membrane voltage at +10mV, see Fig. 1. TTX application suppresses V-dependent Na+ channels and therefore stops all neurotransmission. We show the traces upon TTX to show that currents we were recording prior to TTX application were of synaptic origin, and not due to accidental expression of opsin in the patched cell. Also, this ensures that any current visible after 4AP application is due to monosynaptic transmission and not to a failure of TTX application.

After recording and light stimulation with TTX, we then add 4AP, which is a blocker of presynaptic K+ channels. This prevents the repolarization of the terminals that would occur in response to opsinmediated local depolarization. 4AP application, therefore, allows local opsin-driven depolarizations to reach the threshold for Ca2+-dependent vesicle docking and release. This procedure selectively reveals or unmasks the monosynaptic currents because any non-monosynaptically connected neuron would still need V-dependent Na+ channels to eXectively produce indirect neurotransmission onto the patched cell. The TTX and 4AP application is the gold-standard of opsinassisted synaptic circuit interrogation, pioneered by the Svoboda lab in 2009 and widely used to assess monosynaptic connectivity in multiple brain circuits, as summarized in a recent review (Linders et al., 2022). We now include 2 more sentences near the beginning of the Results to clarify this process and directly point to the Linders review for researchers wanting a deeper explanation of this technique.

The double stimulation is unrelated to our testing of monosynaptic connections. We originally conducted the experiments by delivering 2 pulses of light separated by 50ms, a common way to examine the pair-pulse ratio (PPR) – a physiological measure which is used to probe synapses for short-term plasticity and release probability. However, through discussions with colleagues we realized that the slow decay time of eGtACR1 may complicate interpretation of the response to the second light pulse. Thus, we elected to not report these results and indicated this in the Methods section: “We calculated the paired-pulse ratio (PPR) as the amplitude of the second peak divided by the amplitude of the first peak elicited by the twin stimuli, however due to slow kinetics of eGtACR1 the results would be diPicult to interpret, and therefore we are not currently reporting them.”

(2) Suggestions for improving summary figures:Summary Figure 1a: The circuit diagram (schematic to the right of 1a) is OK but I initially found it a bit diXicult to interpret. For example, it is not clear why pink RA projecting neurons don't reach as far to the right as X or Av projecting neurons, suggesting that they are not really projection neurons. Also, the big question marks in the intermediate zone are not entirely intuitive. It seems there might be a better way of representing this. It might also be worth stating in the figure legend that the interconnectivity patterns shown in the figure between PNs in HVC are based on specific prior studies.

We thank the reviewer for the constructive criticism. We have modified the figure to extend the RA projection line and mentioned in the figure legend that connectivity between PNs is based on prior studies.

Summary Figure 1a: I am not sure I love this figure. There are a few minor issues. First, there are too many browns [Nif/AV and mMAN] which makes it more challenging to clearly disambiguate the diXerent projections. Second, it is unclear why this figure does not represent projections from RA to HVC. My biggest concern with this figure is that it oversimplifies some of the findings. From the figure, one gets the impression that Uva only projects to RA-PNs and that Av only projects to X-PNs even though the authors show connections to other PNs. With the small sample size in this current study for each projection and each PN type, one really cannot rule out that these "minority" projections are not important. I, therefore, suggest that the authors qualitatively represent the strength/probability of connections by weighting with thickness of aXerent connections.

We assume the reviewer is commenting on our summary figure panel 7B. We agree with the referee that this is a simplified representation of our findings. We had indeed indicated in the legend that this was just a “Schematic of the HVC aXerent connectivity map resulting from the present work” and that “For conceptualization purposes, aXerent connectivity to HVC-PNs is shown only when the rate of monosynaptic connectivity reaches 50% of neurons examined”. We have added a title to highlight that this is but a simplification. We have now adjusted the colors to make the figure easier to follow. Based on the reviewers critique we searched for a better method for summarizing the complex connectivity patterns described in this research. We settled on a Sankey diagram of connectivity. This is now Figure 7C. In this diagram, we are able to show the proportion of connections from each input pathway onto each class of neuron and if these connections are poly or monosynaptic. We find this to a straightforward way of displaying all of the connectivity patterns identified in our figure 2-3 and 4-5 look forward to understanding if the reviewers find this a useful way of illustrating our findings.

Minor points:(1) Line 50 - typo - song circuits.

Thank you for catching this.

(2) Line 106 - 111 - The findings suggest that 100% of Uva projections onto HVCRA neurons are monosynaptic. However, because the authors only tested 6 neurons their statements that their findings are so diXerent from other studies, should be somewhat tempered since these other studies (e.g. Moll et al.) looked at 251 neurons in HVC and sampling bias could still somewhat explain the diXerence.

We observed oEPSCs in 43 of 51 (84.3%) HVC-RA neurons recorded (mean rise time = 2.4 ms) and monosynaptic connections onto 100% of the HVC-RA neurons tested (n = 6). Moll et al. combined electrical stimulation of Uva with two-photon calcium imaging (GCaMP6s) of putative HVC-RA neurons (n = 251 neurons). We should note that these are putative HVC-RA neurons because they were not visually identified using retrograde tracing or using some other molecular handle. They report that only ~16% of HVC-RA neurons showed reliable calcium responses following Uva stimulation. Although the experiments by Moll et al are technically impressive, calcium imaging is an insensitive technique for measuring post-synaptic responses, particularly subthreshold responses, when compared to whole-cell patch-clamp recordings. This approach cannot identify monosynaptic connections and is likely limited to only be sensitive suprathreshold activity that likely relies on recruitment of other polysynaptic inputs onto the neurons in HVC. Furthermore, as indicated in the Discussion, our opsin-mediated synaptic interrogation recruits any eGtACR1+ Uva terminal in the slice and therefore will have great likelihood of revealing any existing connections.

A limitation of whole-cell patch-clamp recordings is that it is a laborious low throughput technique. Future experiments using better imaging approaches, like voltage imaging, may be able to weigh in on diXerences between what we report here using whole-cell patch-clamp recordings from visually identified HVC-RA neurons combined with optogenetic manipulations of Uva terminals and the calcium imaging results reported by Moll. Nonetheless, whole-cell patch-clamp recordings combined with optogenetic manipulations is likely to remain the most sensitive method for identifying synaptic connectivity.

(3) Figure 2G - the significance of white circles is not clear.

The figure legend indicates that those highlight and mark the position of “retrogradely labeled HVCprojecting neurons in Uva (cyan, white circles)” to facilitate identification of colocalization with the in-situ markers.

(4) Line 135 - Cardin et al. (J. Neurophys. 2004) is the first to show that song production does not require Nif.

We thank the reviewer pointing this out and we have cited this important study.

(5) Line 183 - This is a confusing sentence because I initially thought that mMAN-mMANHVC PNs was a category!

We switched the dash with a colon.

(6) Figure 4d could use some arrows to identify what is shown. It is assumed that the box represents mMAN. Should it be assumed that Av is not in the plane of this section? If not, this should be stated in the legend. It is also unclear where the anterograde projections are. Is this the dork highway that goes from the box to the dorsal surface? If yes this should be indicated but it should also be made clear why the projections go both in the dorsal as well as the ventral directions.

The inset, as indicated by the lines around it, is a magnification of the terminal fields in Av. We added an explanation of the inset.

(7) Discussion. In the introduction, the authors mention projections from RA to HVC but never end up studying them in the current manuscript which seems like a missed opportunity and perhaps even a weakness of the study. In the discussion, it would certainly be good for the authors to at least discuss the possible significance of these projections and perhaps why they decided not to study them.

We thank the reviewer for the comment. Unfortunately, we couldn’t reliably evoke interpretable currents from RA, and we elected to publish the current version of the paper with these 4 major inputs. Nonetheless, we have indicated in the Introduction and in the Discussion that more inputs (e.g. RA, A11, NCM) remain to be evaluated.

(8) Line 622 - Is this reference incomplete?

We thank the reviewer. We have corrected the reference.

Ben-Tov, M., F. Duarte and R. Mooney (2023). "A neural hub for holistic courtship displays." Curr Biol 33(9): 1640-1653 e1645.Bliss, T. V. and A. R. Gardner-Medwin (1973). "Long-lasting potentiation of synaptic transmission in the dentate area of the unanaestetized rabbit following stimulation of the perforant path." J Physiol 232(2): 357-374.Bliss, T. V. and T. Lomo (1973). "Long-lasting potentiation of synaptic transmission in the dentate area of the anaesthetized rabbit following stimulation of the perforant path." J Physiol 232(2): 331-356.Foster, E. F. and S. W. Bottjer (2001). "Lesions of a telencephalic nucleus in male zebra finches: Influences on vocal behavior in juveniles and adults." J Neurobiol 46(2): 142-165.Koparkar, A., T. L. Warren, J. D. Charlesworth, S. Shin, M. S. Brainard and L. Veit (2024). "Lesions in a songbird vocal circuit increase variability in song syntax." Elife 13.Linders, L. E., L. F. Supiot, W. Du, R. D'Angelo, R. A. H. Adan, D. Riga and F. J. Meye (2022). "Studying Synaptic Connectivity and Strength with Optogenetics and Patch-Clamp Electrophysiology." Int J Mol Sci 23(19).Liu, H. N., T. Kurotani, M. Ren, K. Yamada, Y. Yoshimura and Y. Komatsu (2004). "Presynaptic activity and Ca2+ entry are required for the maintenance of NMDA receptor-independent LTP at visual cortical excitatory synapses." J Neurophysiol 92(2): 1077-1087.Louder, M. I. M., M. Kuroda, D. Taniguchi, J. A. Komorowska-Muller, Y. Morohashi, M. Takahashi, M. Sanchez-Valpuesta, K. Wada, Y. Okada, H. Hioki and Y. Yazaki-Sugiyama (2024). "Transient sensorimotor projections in the developmental song learning period." Cell Rep 43(5): 114196.Pastalkova, E., P. Serrano, D. Pinkhasova, E. Wallace, A. A. Fenton and T. C. Sacktor (2006). "Storage of spatial information by the maintenance mechanism of LTP." Science 313(5790): 1141-1144.Petreanu, L., T. Mao, S. M. Sternson and K. Svoboda (2009). "The subcellular organization of neocortical excitatory connections." Nature 457(7233): 1142-1145.Roberts, T. F., S. M. Gobes, M. Murugan, B. P. Olveczky and R. Mooney (2012). "Motor circuits are required to encode a sensory model for imitative learning." Nat Neurosci 15(10): 1454-1459.Roberts, T. F., E. Hisey, M. Tanaka, M. G. Kearney, G. Chattree, C. F. Yang, N. M. Shah and R. Mooney (2017). "Identification of a motor-to-auditory pathway important for vocal learning." Nat Neurosci 20(7): 978-986.Roberts, T. F., M. E. Klein, M. F. Kubke, J. M. Wild and R. Mooney (2008). "Telencephalic neurons monosynaptically link brainstem and forebrain premotor networks necessary for song." J Neurosci 28(13): 3479-3489.Trusel, M., A. Cavaccini, M. Gritti, B. Greco, P. P. Saintot, C. Nazzaro, M. Cerovic, I. Morella, R. Brambilla and R. Tonini (2015). "Coordinated Regulation of Synaptic Plasticity at Striatopallidal and Striatonigral Neurons Orchestrates Motor Control." Cell Rep 13(7): 1353-1365.Trusel, M., A. Nuno-Perez, S. Lecca, H. Harada, A. L. Lalive, M. Congiu, K. Takemoto, T. Takahashi, F. Ferraguti and M. Mameli (2019). "Punishment-Predictive Cues Guide Avoidance through Potentiation of Hypothalamus-to-Habenula Synapses." Neuron 102(1): 120-127.e124.Vates, G. E., B. M. Broome, C. V. Mello and F. Nottebohm (1996). "Auditory pathways of caudal telencephalon and their relation to the song system of adult male zebra finches." Journal of Comparative Neurology 366(4): 613-642.Xu, T., X. Yu, A. J. Perlik, W. F. Tobin, J. A. Zweig, K. Tennant, T. Jones and Y. Zuo (2009). "Rapid formation and selective stabilization of synapses for enduring motor memories." Nature 462(7275): 915-919.Zhao, W., F. Garcia-Oscos, D. Dinh and T. F. Roberts (2019). "Inception of memories that guide vocal learning in the songbird." Science 366: 83 - 89.